# Accelerated evolution in networked metapopulations of *Pseudomonas aeruginosa*

**Partha Pratim Chakraborty[1†], Rees Kassen[1,2]***

[1]University of Ottawa, Ottawa, Canada; [2]McGill University, Montreal, Canada

**Abstract** Natural populations are often spatially structured, meaning they exist as metapopulations composed of subpopulations connected by migration. Little is known about the impact of spatial structure, in particular the topology of connections among subpopulations, on adaptive evolution. Typically, spatial structure slows adaptation, although some models suggest topologies that concentrate dispersing individuals through a central hub can accelerate adaptation above that of a well-mixed system. We provide evidence to support this claim and show acceleration is accompanied by high rates of parallel evolution. Our results suggest metapopulation topology can be a potent force driving evolutionary dynamics and patterns of genomic repeatability in structured landscapes such as those involving the spread of pathogens or invasive species.

## Editor's evaluation

This manuscript shows that certain forms of population subdivision (in particular "start" topologies) can accelerate adaptation, representing an important empirical confirmation of general predictions from evolutionary graph theory. The experiments and analysis are rigorous and provide convincing evidence for the main claims of the paper. There is some ambiguity and potential disagreement about precisely how these results relate to previous theoretical expectations, but these are matters of interpretation and we believe this article should stimulate future theory work that will further clarify the role of population structure on adaptation, particularly in situations more directly comparable to experimental settings.

*For correspondence:
rees.kassen@mcgill.ca

Present address: †Gulbenkian Institute for Molecular Medicine, Rua da Quinta Grande, Oeiras, Portugal

**Competing interest:** The authors declare that no competing interests exist.

## Introduction

Natural populations are often spatially structured as metapopulations, meaning they are geographically subdivided and connected by migration. Examples include infectious diseases that transmit through contact networks among hosts (*Leventhal et al., 2015*), species ranges that can be discontiguous, especially at range edges where peripheral populations are connected via dispersal to a more well-connected central population (*Sagarin et al., 2006*; *Pennington et al., 2021*), and patients (along with their colonizing pathogens) that move among wards in a hospital (*Myall et al., 2021*). How metapopulation topology, the arrangement of connections among subpopulations, influences the dynamics of adaptation is not well understood. The central issues concern the impact of topology on rates of adaptation, the time to generate and fix beneficial mutations (*Tkadlec et al., 2019*; *Yagoobi and Traulsen, 2021*; *Frean et al., 2013*), and the population genetic mechanisms responsible.

The conventional view, based on a model of reproduction involving offspring replacing parents *en masse* in non-overlapping generations and leading to global competition among individuals, is that the topology of connections among subpopulations has little influence on rates of adaptation (*Guillaume, 2011*; *Maruyama, 1970*; *Slatkin, 1981*; *Slatkin, 1985*). Migration in these models

tends to slow but not prevent adaptation across a metapopulation relative to a well-mixed, fully connected system when the rate of selection, $s$, exceeds that of migration, $m$. Rates of adaptation in spatially structured populations can be slowed further by clonal interference, the competition for fixation among independently arising beneficial mutations (*Gordo and Campos, 2006*), an effect that can be particularly strong in large populations because mutation supply rates, being the product of population size, $N$, and the genome-wide mutation rate, $U$, are often high in large populations.

By contrast, topology can strongly modulate the dynamics of adaptation in finite populations where individuals reproduce in overlapping generations and compete locally with each other, either decreasing or increasing rates of adaptation relative to a well-mixed system depending on the arrangement of subpopulations (*Yagoobi and Traulsen, 2021*; *Lieberman et al., 2005*; *Pavlogiannis et al., 2018*; *Tkadlec et al., 2021*; *Hindersin and Traulsen, 2015*). Much attention has been paid to how 'star' topologies, comprised of a central 'hub' connected to peripheral 'leaf' subpopulations, can increase the probability of fixation for beneficial mutations above that expected for a well-mixed population because this result is unexpected from standard models (*Maruyama, 1970*; *Slatkin, 1981*) and may reasonably capture features of many real-world scenarios such as contact networks of infectious diseases, the edge of species ranges, or patient transfers between regional care centres and large urban hospitals (*Donker et al., 2017*; *Nekkab et al., 2020*). Rates of adaptation, measured as the establishment time for beneficial mutations destined to fix, are usually slower in stars than well-mixed populations (*Frean et al., 2013*) but can be accelerated when mutation supply rates in each subpopulation (which is $NU$, as defined above, times $m$, the migration rate into a subpopulation) are <1 (*Tkadlec et al., 2019*; *Yagoobi and Traulsen, 2021*), possibly because rare beneficial mutations become concentrated in the hub and so are less likely to be stochastically lost when rare, a phenomenon called drift loss.

Empirical evidence on the impact of topology on rates of adaptation is limited. In microbial evolution experiments where population sizes are typically very large, various forms of spatial structure usually slow down adaptation relative to a well-mixed condition (*Miralles et al., 1999*; *Habets et al., 2006*; *Habets et al., 2007*; *Perfeito et al., 2008*; *Kryazhimskiy et al., 2012*; *Bailey, 2021*). The one exception is a study where faster adaptation occurred in structured compared to unstructured populations, a result attributed to the ability of the structured population to explore higher fitness peaks in a rugged fitness landscape (*Nahum et al., 2015*). The only study to evaluate the impact of topology per se on adaptation across a range of population sizes tracked an initially rare beneficial mutation introduced into one peripheral 'leaf' through both 'star' and well-mixed metapopulations (*Chakraborty et al., 2023*). The mutation spread more rapidly through star metapopulations than well-mixed populations when migration rates were low ($m < 0.01\%$) and this difference could be exaggerated by reducing population sizes (from $N \sim 10^7$ to $10^5$ CFU/ml) and biasing migration from leaves towards the hub. These results are consistent with a mechanism for acceleration involving a reduced probability of drift loss in stars: beneficial mutations rising to high frequency in their introduced subpopulation become concentrated in the hub via migration, allowing them to spread rapidly to other leaves (*Chakraborty et al., 2023*).

Is acceleration possible under more prolonged and open-ended evolution, where mutations arise naturally at any location across the metapopulation and compete with each other for fixation through clonal interference? Evolutionary graph theory, which is grounded in a model of reproduction that generates local competition among individuals, suggests the answer depends on the extent of clonal interference associated with migration (*Frean et al., 2013*; *Sharma et al., 2023*). Only when clonal interference is sufficiently low to allow single, initially rare beneficial mutations to gain access to the hub do we expect to see acceleration (*Tkadlec et al., 2019*; *Yagoobi and Traulsen, 2021*; *Sharma et al., 2023*). To test this prediction, we tracked rates of adaptation of the opportunistic pathogen *Pseudomonas aeruginosa* (PA14) in the presence of subinhibitory concentrations of the fluoroquinolone antibiotic ciprofloxacin for approximately 100 generations in 4-patch star and well-mixed populations across a range of mutation supply rates. We manipulate mutation supply rates in each subpopulation by adjusting both the effective population size and migration rate (*Figure 1*; see Materials and methods). Each treatment combination is replicated eight times.

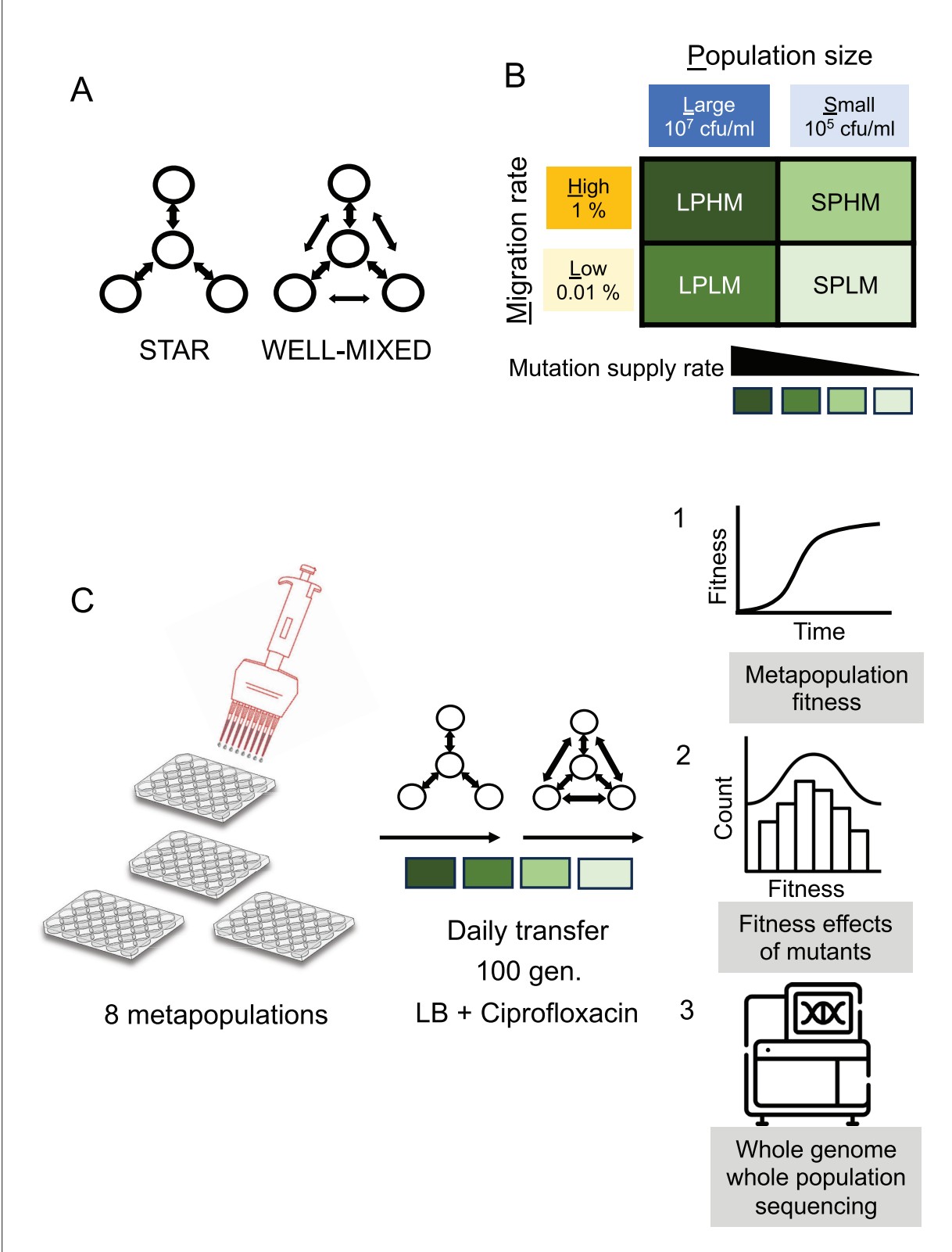

**Figure 1.** Design for de novo evolution experiment. (**A**) Two topologies, star or well-mixed, constructed among four subpopulations. Arrows depict dispersal routes among subpopulations (circles). (**B**) Four combinations of mutation supply rates achieved by manipulating both effective population sizes of the subpopulations and the migration rates among the subpopulations. (**C**) Experimental evolution setup and subsequent assays performed.

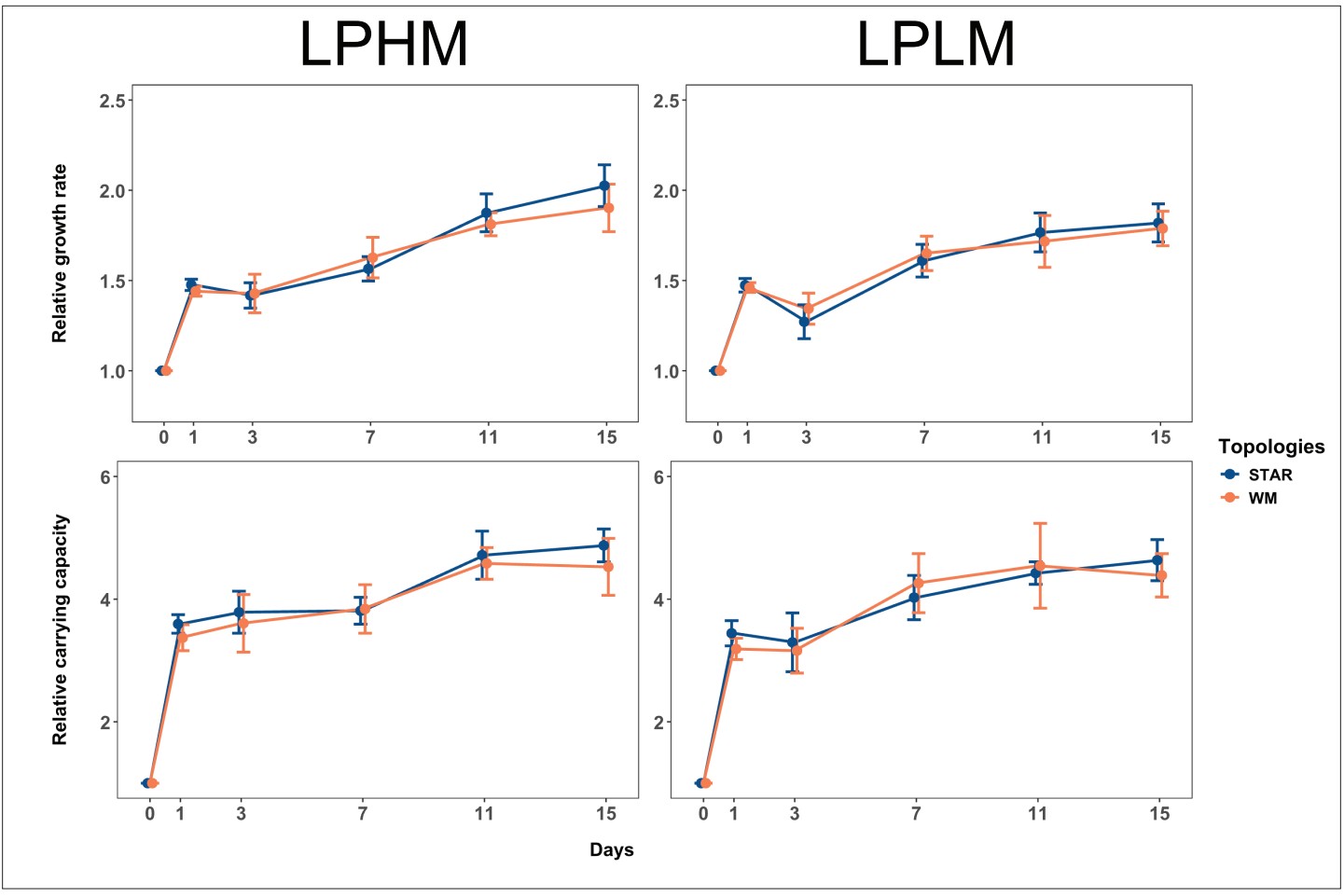

**Figure 2.** Dynamics of adaptation in large metapopulations. Fitness trajectories in large metapopulations connected by high and low migration rates (LPHM and LPLM). Increase in relative growth rate (upper panel) and relative carrying capacity (lower panel) in metapopulations propagated by either the star or the well-mixed topology with the LPHM (left panel) or LPLM (right panel) regime over the experimental time-period. Each point is the mean of eight replicate metapopulations for a particular day and network topology; error bars show 1 standard error of the mean (SE). Raw data from each replicate metapopulation shown in **Figure 2—figure supplement 1**. For large metapopulations (LPHM and LPLM), approximately 6.67 generations of growth happened per transfer.

The online version of this article includes the following figure supplement(s) for figure 2:

**Figure supplement 1.** Dynamics of adaptation in large metapopulations with individual replicate metapopulations.

## Results

Our results are shown in **Figures 2 and 3**. Fitness, measured as both growth rate ($r$) and stationary phase density ($K$) relative to the ancestral strain, was faster in star metapopulations relative to well-mixed populations when effective population sizes were small (main effect of network, p = 0.06 and p < 0.01 for relative $r$ and $K$, respectively; effect sizes (Cohen's $d$ for the difference between star and well-mixed treatments) $r$ = 0.929, 95% CI: [−0.035 to 1.89]; $K$ = 1.53, 95% CI: [0.586 to 2.47]), irrespective of migration rate (relative $r$: p = 0.73 and relative $K$: p = 0.97) and even after accounting for the removal of outliers (**Figure 3—figure supplement 2**). Larger metapopulations adapted faster than small ones, as expected (**Wilke, 2004**; **Orr, 2000**), but there was no effect of topology on rates of fitness increase at large effective population sizes (main effect of network treatment for relative $r$: p = 0.95 and relative $K$: p = 0.60) across the two migration rates (relative $r$: p = 0.42 and relative $K$: p = 0.51). These results demonstrate that star topologies can accelerate adaptation by as much as 1.5x over that of well-mixed metapopulations when mutation supply is low but not high.

We previously showed that a single beneficial mutation spreads more rapidly in stars than well-mixed populations when selection is strong relative to migration because a rare beneficial mutant

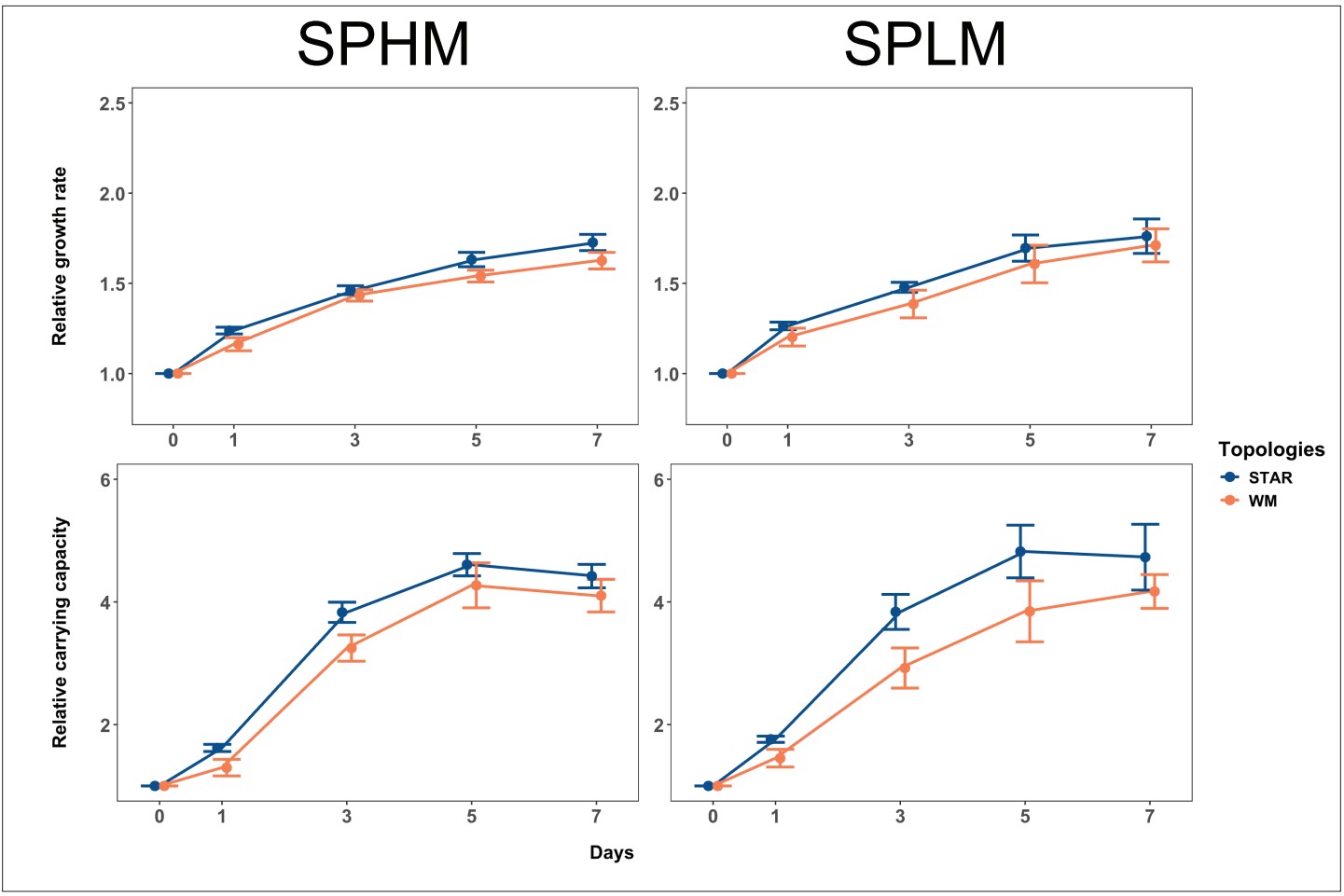

**Figure 3.** Dynamics of adaptation in small metapopulations. Fitness trajectories in small metapopulations (SPHM and SPLM) connected by high and low migration rates. Increase in relative growth rate (upper panel) and relative carrying capacity (lower panel) in metapopulations propagated by either the star or the well-mixed topology with the SPHM (left vertical panel) or SPLM (right vertical panel) regime over the experimental time-period. Each point is the mean of eight replicate metapopulations for a particular day and network topology; error bars show 1 standard error of the mean (SE). Raw data from each replicate metapopulation is shown as faded lines in **Figure 3—figure supplement 1**. Small metapopulations (SPHM and SPLM) experienced approximately 13.28 generations of growth per transfer.

The online version of this article includes the following figure supplement(s) for figure 3:

**Figure supplement 1.** Dynamics of adaptation in small metapopulations with individual replicate metapopulations.

**Figure supplement 2.** Detection of outliers in the SPHM and SPLM dataset.

is less likely to be lost due to drift in a newly colonized patch, consistent with the first mechanism (*Chakraborty et al., 2023*). Establishment times, being the time required to substitute a beneficial mutation that is destined to fix (*Frean et al., 2013*), may also be faster in stars than well-mixed populations because beneficial mutants experience reduced clonal interference owing to the fact that leaf subpopulations in stars exchange migrants only with the hub whereas all subpopulations – leaves and hub – share migrants equally in a well-mixed metapopulation. Reduced clonal interference could allow beneficial mutants arising in a leaf both to fix more rapidly there and to seed other leaves via the hub (*Kuo et al., 2025*). Adaptation could be further accelerated if reduced clonal interference in stars is accompanied by a lower probability of drift loss for rare beneficial mutants colonizing a new patch.

To evaluate these hypotheses, we first checked that clonal interference was in fact lower in stars than well-mixed metapopulations. We estimated the number and spectrum of mutations segregating at the end of our experiment by sequencing 24 evolved metapopulations (6 for each population size x topology combination). We achieved ~3000-fold sequencing depth and ~98.5% genome-wide coverage for each metapopulation which, after comparing the data with ancestral PA14 genomes (PA14 and PA14-LacZ), allowed us to detect all mutations segregating above ~5% frequency in each

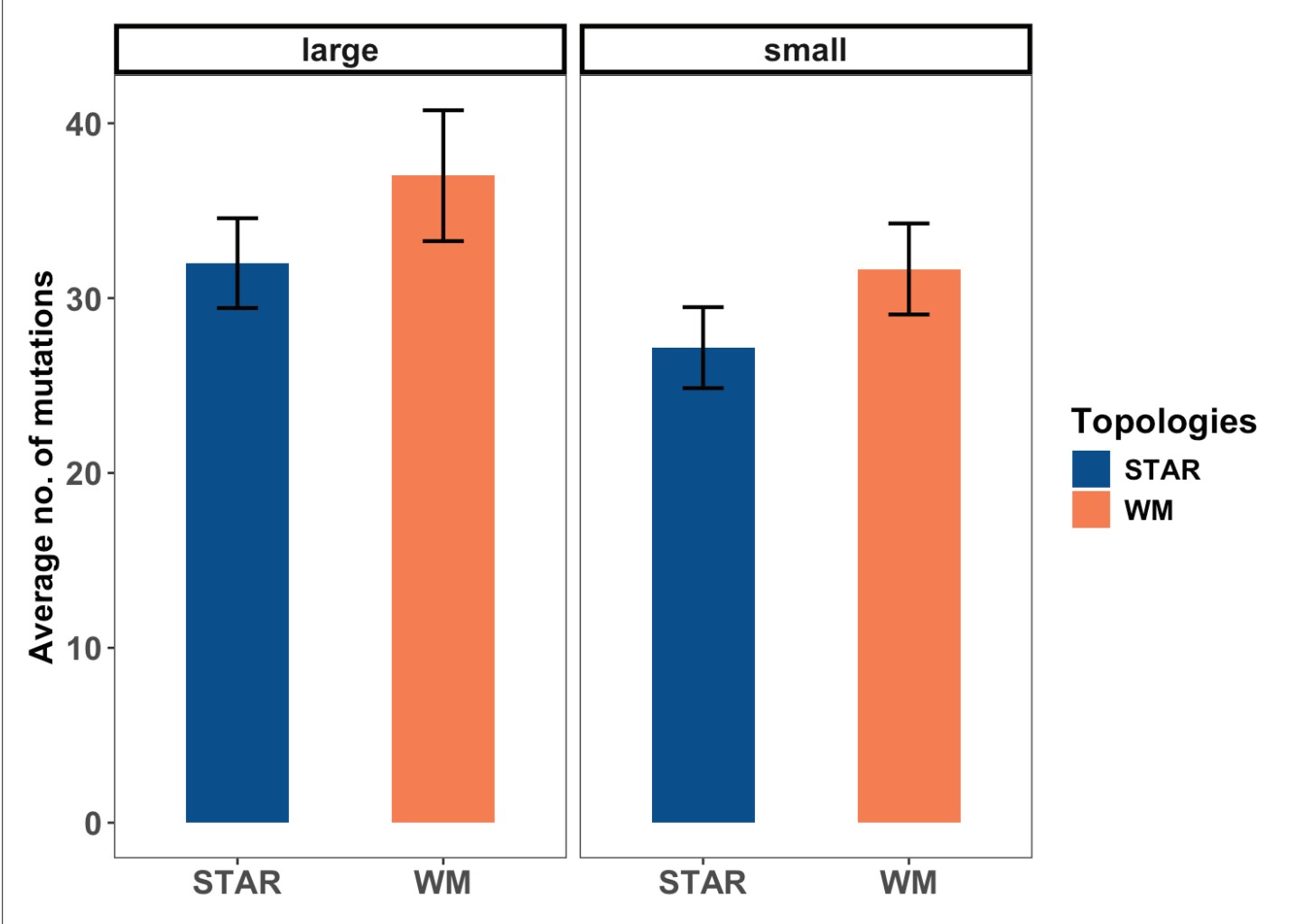

**Figure 4.** Number of mutational changes in each population size/network topology treatment combinations. Shown are the mean number of mutational changes for each network topology treatment (star and well-mixed) under each population size (large and small). Error bars represent 1 standard error of the mean.

The online version of this article includes the following figure supplement(s) for figure 4:

**Figure supplement 1.** Average frequency of adaptive mutations in the metapopulations.

**Figure supplement 2.** Average frequency of mutations segregating in networks with different population sizes.

metapopulation. *Figure 4* shows the average number of mutations segregating in each treatment. The results indicate that while all metapopulations in our experiment evolve in the clonal interference regime, large populations harbour more mutations on average than small populations (two-way ANOVA, main effect population size, p = 0.09), and well-mixed metapopulations support more mutations than star populations at both population sizes (two-way ANOVA, main effect of network treatment, p = 0.11, effect size of the difference between star and well-mixed mean averaged over both population sizes –0.678, 95% CI: [−1.56 to 0.203]). While these effects are not formally significant at the conventional level (p < 0.05), they reassure us that our experimental manipulations modulated the level of clonal interference in the expected directions.

Reduced clonal interference could allow large effect beneficial mutants to increase in frequency more rapidly in stars relative to a well-mixed metapopulation. If these beneficial mutants are also less likely to be lost to drift during colonization of a new patch, large effect beneficial mutants should be enriched in low population size stars where we see accelerated adaptation. To test this prediction, we collected 48 clones from the last timepoint of each of the evolved metapopulations and assayed stationary phase density at 24 hr, *K*, as a proxy for fitness in the selection medium (*Figure 5*). Large

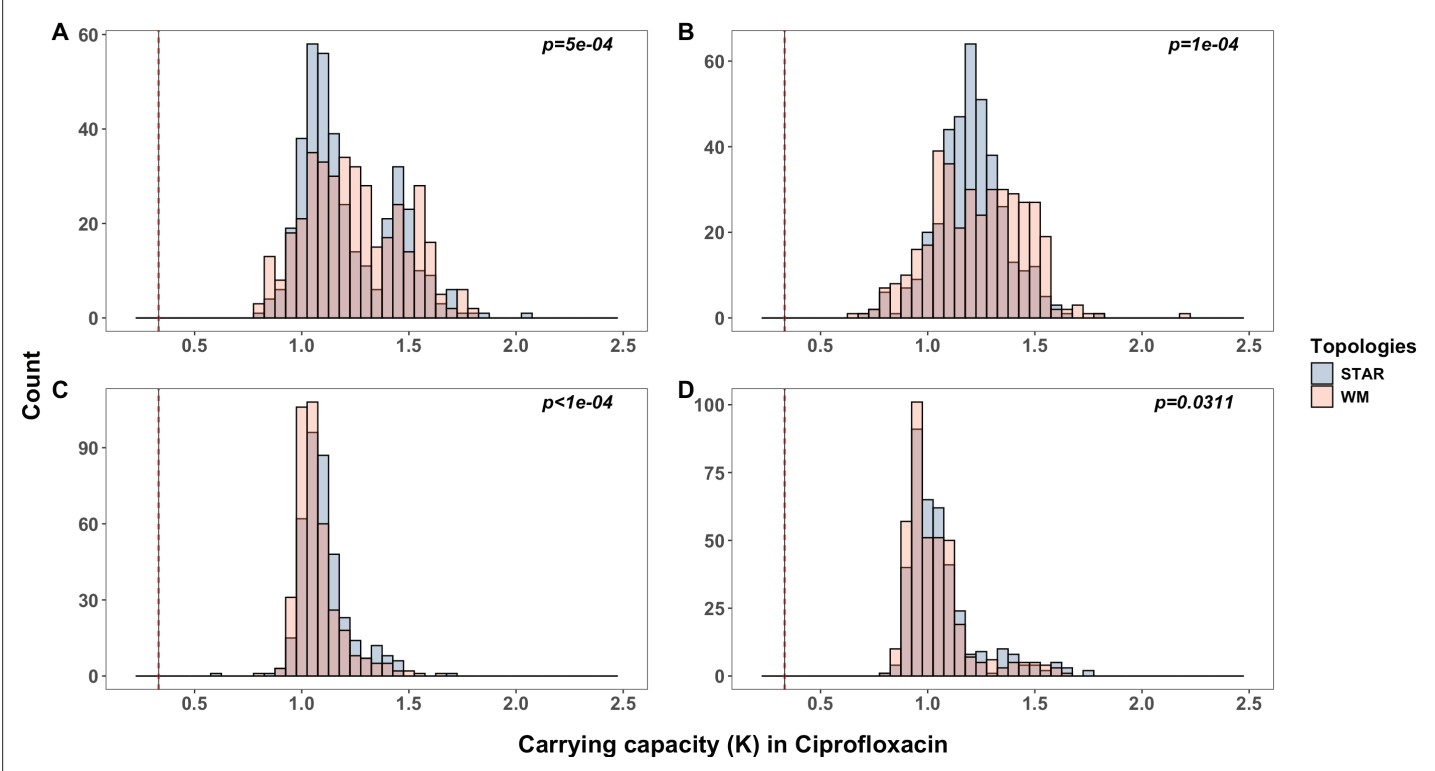

**Figure 5.** Distribution of fitness effects among isolates in the selection medium. Absolute fitness of evolved end-point isolates in the selection medium (LB supplemented with subinhibitory concentration of ciprofloxacin). Fitness effect distributions are shown as histograms of carrying capacity (*K*) for all mutants isolated from the large (LPHM and LPLM) metapopulations (**A** and **B**, respectively) or the small (SPHM and SPLM) metapopulations (**C** and **D**, respectively). Blue and red bars denote isolates collected from metapopulations propagated either by the star or the well-mixed topologies, respectively. Statistical significance determined at p < 0.05 by permutation K–S test (10,000 permutations). The vertical line on the left-hand side of each histogram is the mean *K* of the ancestor in the selection medium and the red dotted vertical lines are 1 standard error (SE) of the mean.

The online version of this article includes the following figure supplement(s) for figure 5:

**Figure supplement 1.** Empirical cumulative distribution function (ECDF) plots for observed absolute fitnesses of evolved clones.

effect clones are preferentially fixed in the well-mixed population compared to the star when population sizes are large (*Figure 4*, panels A, B, permutation K–S test, $p < 10^{-3}$ and $p < 10^{-3}$, for high and low migration rates, respectively, for 10,000 permutations), consistent with strong clonal interference biasing fixation towards large effect mutants. In small populations, by contrast, star metapopulations harboured more large effect clones compared to the well-mixed metapopulations across both high and low migration rates (*Figure 4*, panels C, D, permutation K–S test, $p < 10^{-3}$ and $p < 0.05$, for high and low migration rates, respectively, for 10,000 permutations). These results are further reinforced by inspecting the empirical cumulative distribution functions for *K*, which are right-shifted at small effective population sizes reflecting the predominance of larger effect mutants in stars relative to well-mixed populations (*Figure 5—figure supplement 1*).

If the cause of acceleration is the preferential substitution of large effect mutations, then we should also expect to see higher levels of repeated, or parallel, evolution in small *N* stars than in any other treatment of our experiment. Parallel evolution is often taken to be an indicator of strong selection acting in a particular locus, as it is unlikely that the same genes or mutations would fix repeatedly by chance alone (*Orr, 2005*). Equally, it could be due to preferential substitution caused by the reduced probability of drift loss, and so higher fixation probability, among the rarer class of larger effect beneficial mutations available to selection (*Orr, 2005*; *Chevin et al., 2010*; *Gillespie, 1984*; *Kassen and Bataillon, 2006*; *Fisher, 1930*) together with the more rapid spread facilitated by reduced clonal interference. We tested this prediction by using our sequencing results to calculate the rate of repeated, or parallel, evolution for each mutated gene in our data set. We find that parallel evolution was higher in the small *N* star metapopulations compared with all others, an observation confirmed by the significant

**Table 1.** Population-level parallelism.

Difference in population-level parallelism between the two topologies for each effective population size. Differences between the mean values for each metric (estimate) in the star and the well-mixed topologies are presented. Low dispersion and Jaccard index values, and high $C$-scores, correspond to high rates of parallelism. A positive value denotes the calculated metric for the star topology is higher than the well-mixed topology and vice versa. Significance ($p < 0.05$) is determined by a two-way ANOVA for dispersion and a two-way ANOVA followed by a permutation test (10,000 permutations) for Jaccard distance and $C$-score (Materials and methods).

| | Parallelism estimate = (Star − Well-mixed) | | | | | |
|---|---|---|---|---|---|---|
| | Metric | | | | | |
| Population size | Dispersion | | Jaccard distance | | $C$-score | |
| | Estimate | Significance | Estimate | Significance | Estimate | Significance |
| Large | 0.0167 | 0.7153 | 0.0165 | 0.6569 | 0.0219 | 0.5356 |
| Small | −0.1265 | **0.0108** | −0.1514 | **0.0009** | 2.0156 | **0.0029** |

interaction between population size and network topology for three metrics of parallelism ($p < 0.05$; see methods). Closer inspection reveals star and well-mixed topologies do not differ in the extent of parallelism in large metapopulations but they do for small metapopulations, stars showing significantly higher parallelism than their well-mixed counterparts (*Table 1*), a result also reflected by higher frequency of all mutations (those appearing repeatedly and as singletons) in small stars compared to the small well-mixed metapopulations (*Figure 4—figure supplement 2*). As there is little reason to suspect the distribution of fitness effects among beneficial mutations available to selection differs systematically between star and well-mixed topologies for a given population size, the high levels of parallelism in the small $N$ star must be associated with a higher fixation probability among the first mutations arising in the experiment. Given the over-representation of large effect beneficial mutants in small $N$ star metapopulations, these results suggest accelerated adaptation is due to the combined effect of reduced clonal interference and large effect beneficial mutations having a higher probability of fixation because they avoid drift loss when rare in stars compared to well-mixed populations.

As expected from the strong selection pressure exerted by ciprofloxacin, we found many known resistance mutations circulating in moderate to high frequencies in these metapopulations (*Figure 6*). We identified frequent mutations (small indels and nonsynonymous SNPs) in the negative regulators and components of efflux pumps such as *mexA* and *mexR* (mexAB-oprM), *mexS* (MexEF-OprN), and *nfxB* (MexCD-OprJ) that are constitutively expressed when it is necessary to decrease intracellular concentration of ciprofloxacin (*Poole, 2005*; *Richardot et al., 2016*). We also uncovered mutations that prevent ciprofloxacin from binding to the DNA-modifying subunits of DNA gyrases (gyrA and gyrB) (*Yoshida et al., 1990a*; *Yoshida et al., 1990b*; *Barnard and Maxwell, 2001*). Not unexpectedly, our evolved metapopulations also harbour mutations in genes that do not confer resistance to ciprofloxacin but are global regulators of quorum sensing (*lasR*), c-di-GMP signalling and biofilm formation (*wspA/F/R*, *morA*), virulence and twitching motility (*pilB/C/D/F*) (*Rumbaugh et al., 2000*; *Schuster and Greenberg, 2006*; *Williams and Cámara, 2009*; *Moradali et al., 2017*; *Chadha et al., 2022*; *Choy et al., 2004*; *Ha et al., 2014*; *Harrison et al., 2020*; *Katharios-Lanwermeyer et al., 2021*; *Ryder et al., 2007*; *Römling et al., 2013*; *Ha and O'Toole, 2015*; *Jenal et al., 2017*; *Ma et al., 2022*). We and others commonly observe these same mutations in rich laboratory growth media, where their fitness advantage is thought to be associated with reducing the costs of metabolism (*Mould et al., 2022*; *Schick et al., 2022*).

A last striking result from our genomic analyses is that metapopulations evolved under high and low mutation supply diverge in the repertoire of resistance mutations they accumulate over the evolutionary period. Notable examples include the enrichment of *mexS* and *gyrA* mutations exclusively in small and large metapopulations, respectively (binomial test, $p < 0.005$ and $p < 0.05$, respectively). We see the same pattern for putative non-resistance mutations as well: *lasR* and *wspA* being mutated in the large (binomial test, $p < 0.005$ and $p < 0.05$, for lasR and wspA, respectively) but not in the small metapopulations. Although we do not uncover any mutations that are network specific in the large metapopulations, there is a tendency for mutations in *wspA* and *wspF* to be enriched in the

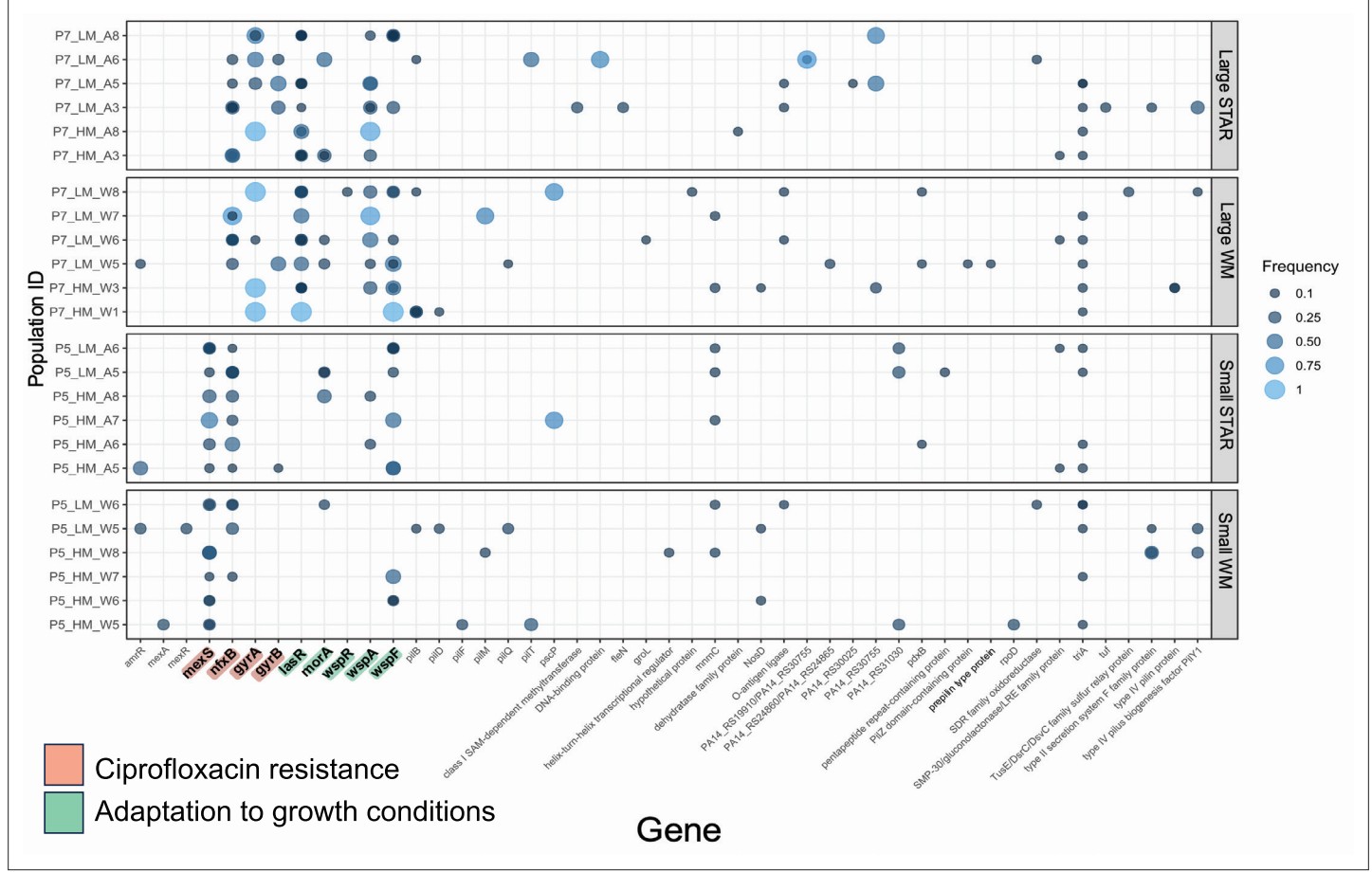

**Figure 6.** Genetic changes detected in the metapopulations (not showing intergenic mutations; full data file is available in **Figure 6—figure supplement 1**) after ~100 generations of evolution in the selection media (LB supplemented with subinhibitory concentration of ciprofloxacin). From the top, the first and the second panels show large star and large well-mixed metapopulations, respectively. Similarly, the third and the fourth panels show small star and small well-mixed metapopulations, respectively. Mutation in genes highlighted with either red or green background denotes recurring canonical ciprofloxacin resistance genes and genes that are relevant for adaptation of *P. aeruginosa* to laboratory conditions, respectively. Size of the circles depicts observed frequencies of mutants in the metapopulations. More than one circle for a gene represents the presence of distinct genetic variants (alleles) in the same metapopulation.

The online version of this article includes the following figure supplement(s) for figure 6:

**Figure supplement 1.** Genetic changes detected in the metapopulations (showing all mutations above the frequency of 8%) after ~100 generations of evolution in the selection media (LB supplemented with subinhibitory concentration of ciprofloxacin).

**Figure supplement 2.** Frequency spectra for all mutations.

small star metapopulations (*wspA* and *wspF* mutations in six star compared to *wspF* mutations in two well-mixed metapopulations). Additionally, it is notable that mutations in *nfxB* are present in all (six out of six) small star but only 50% (three out of six) in the small well-mixed metapopulations sequenced, although the difference is not statistically significant (binomial test, p = 0.08). Furthermore, the average frequency of resistance-conferring *mexS* and *nfxB* mutations is modestly higher in small stars relative to small well-mixed metapopulations (**Figure 4—figure supplement 1**), although this effect is not formally significant (two-sample *t*-test p = 0.1585 and p = 0.2606, for *nfxB* and *mexS*, respectively, and p = 0.1729, Fisher's method of combining multiple probabilities). This result, taken together with the observation that fewer mutations segregate in small stars compared to small well-mixed networks (**Figure 4**), lends further support to the idea that small stars lead to rapid enrichment of beneficial driver mutations.

Our genomic data is also consistent with the idea that mutation supply rate can bias the spectrum of mutations contributing to adaptation. Large populations with high mutation supply rates have access to a larger spectrum of mutations, and clonal interference during substitution can lead

to preferential fixation of those with the largest fitness effects. Interestingly, however, there is little evidence that network topology itself biases the spectrum of mutations available to selection; rather, it is the preferential enrichment of large effect mutations that avoid drift loss in star metapopulations that governs the dynamics of molecular variation in these situations.

## Discussion

Our work provides direct, experimental evidence that the topology of connections in a metapopulation can alter the pace of adaptive evolution and the identity of the mutations responsible. Specifically, we have shown that small $N$ star metapopulations, those characterized by satellite populations connected through a central hub, can accelerate adaptation over and above that observed in a well-mixed, fully connected setting. Acceleration is associated with an enrichment of large effect mutants and results in higher levels of gene-level parallelism compared to well-mixed populations of the same size. These results are consistent with an underlying mechanism involving the concentration of beneficial mutations in the hub by dispersal and facilitating their rapid spread to other leaves in the metapopulation. The contribution of reduced probability of drift loss and lower levels of clonal interference in accelerating adaptation is a topic that deserves further exploration.

That we observed acceleration in the small $N$ stars is surprising because it does not fit with the two main models of reproduction used to track allele frequencies in metapopulations. The approach most common in population genetics, the Fisher–Wright model, assumes reproduction involves bulk replacement of parents by offspring in non-overlapping generations leading to global competition among individuals each generation. Metapopulation topology has little or no effect on fixation probabilities in this model, and the pace of adaptation is almost always fastest in well-mixed populations (*Maruyama, 1970*; *Slatkin, 1981*). An alternative approach, evolutionary graph theory, is based on the Moran model of individual reproduction and death, which leads to overlapping generations and local competition among neighbouring individuals. Moran-based models show that stars can increase the fixation probability of beneficial mutants relative to a well-mixed population, although at the cost of longer fixation times (*Tkadlec et al., 2019*; *Frean et al., 2013*; *Lieberman et al., 2005*). Our results show that both models fail to capture accurately the dynamics of adaptation on stars: in contrast to Fisher–Wright models, stars can adapt faster than well-mixed populations, whereas times to fixation appear to be much smaller than anticipated for the Moran process on graphs. Progress towards a general theory of adaptation in spatially structured populations will thus demand we rethink the central assumptions underpinning reproduction in both models.

Our results also bear on an old debate around how to appropriately conceptualize the role of spatial structure in adaptive evolution. One view, due to *Fisher, 1930*, effectively ignores spatial structure: selection across subpopulations is modelled as if it was occurring in a large, panmictic and freely recombining population, the fitness effect of any given allele being described by its average fitness improvement over the current wild type when tested against all possible genetic backgrounds from all subpopulations. An alternative view, attributed to *Wright, 1932* and called the shifting balance model, treats spatial structure as an integral feature of adaptation: selection occurs independently in small, spatially separated subpopulations connected by migration. The fitness effect of an allele depends both on the genetic background in which it arises in a given subpopulation and on how effectively it outcompetes distinct beneficial mutants originating in other subpopulations. Our experimental results provide insight into the mechanisms driving this second phase of the shifting balance model, the competition among beneficial mutants derived from distinct subpopulations, by showing how, when mutation supply rates are low, the combined effects of reduced clonal interference and higher probability of avoiding drift loss can allow beneficial mutants to spread rapidly through a metapopulation.

How common is accelerated adaptation on star-like structures in more natural systems? This question remains difficult to answer. Many microbial communities occupy environments such as soil or hosts that are spatially structured (*Dang and Lovell, 2016*; *Hall-Stoodley et al., 2004*; *Nadell et al., 2016*; *Kraemer and Boynton, 2017*; *García-Bayona and Comstock, 2018*; *Stacy et al., 2016*; *Hajishengallis et al., 2023*; *Werner et al., 2014*), and some studies have shown that spatial structure following population bottlenecks imposed during host dissociation or biofilm dispersal can have profoundly different outcomes depending on the spatial structure of the population (*Nadell et al., 2016*; *Abel et al., 2015*; *Steenackers et al., 2016*). The fact that we only observed acceleration at

the smallest population sizes in our experiment ($N_e$ ~$10^5$ individuals) suggests there could be an upper threshold of population size above which acceleration is unlikely to be observed. Nevertheless, the metapopulation structure of many large organisms may resemble that of a star, for example when range edge populations are sparse compared to the more abundant centre or due to habitat fragmentation (*Sagarin et al., 2006*; *Pennington et al., 2021*). If so, star topologies could be an explanation for the rapid spread of invasive species. Moreover, many scale-free networks, which often characterize contact networks in epidemiology, contain sub-structures resembling stars and are known to be weak amplifiers of selection (*Leventhal et al., 2015*; *Lieberman et al., 2005*). It is not inconceivable that metapopulation structures such as the star play a more important role in adaptive evolution than has been recognized up to now.

## Materials and methods

### Microbial strains and conditions

For all experiments, clonal populations of *P. aeruginosa* strain 14 (PA14) and PA14:*lacZ*, isogenic to PA14 except with an insertion in the *lacZ* gene, respectively, were used. Colonies possessing the *lacZ* insertion appear blue when cultured on agar plates supplemented with 40 mg/l of 5-bromo-4-chloro-3-indolyl-beta-D-galactopyranoside (X-Gal), and are visually distinct from the PA14 white colouration. The neutrality of the *lacZ* marker was confirmed in our experimental environments by measuring the fitness of the marked strain relative to the unmarked strain. Populations were cultured in 24-well plates with 1.5 ml of media in each well, in an orbital shaker (150 RPM) at 37°C. The culture media consisted of Luria Bertani broth (LB: bacto-tryptone 10 g/l, yeast extract 5 g/l, NaCl 10 g/l) supplemented with 40 ng/ml of the fluoroquinolone antibiotic, ciprofloxacin. This particular subinhibitory concentration of ciprofloxacin was chosen to exert a moderate level of selection that would slow down the dynamics of fixation of resistant mutations compared to a lethal dose above the minimum inhibitory concentration. This approach allows us to discern the effect of network topology on the process of adaptive substitution with weakened interference of strong selection (Supplementary figure 7 in *Chakraborty et al., 2023*). Lower ciprofloxacin concentrations did not enrich resistant mutations to high levels in the populations within the experimental time-period tested (Supplementary figure 8 in *Chakraborty et al., 2023*). All strains and evolving populations were cryopreserved at −80°C in 20% (vol/vol) glycerol.

### Evolution experiment

A single metapopulation consisted of four subpopulations, one subpopulation being located on each of four different 24-well plates. Plate 2 was always assigned as the hub, and plates 1, 3, and 4 were treated as the leaves. Half of the metapopulations (every odd numbered replicate) in this experiment was inoculated with clonal PA14 and the other half (every even numbered replicate) with clonal PA14:*lacZ*. This was necessary in order to track any cross-contamination between metapopulations inhabiting adjacent wells in the 24-well plate.

In a metapopulation, mutation supply rate can be manipulated at two different levels, at the level of subpopulations – which can be manipulated by changing the effective population size ($N_e$) of the constitutive subpopulations and at the level of the whole metapopulation – by modifying the rate of migration (*m*) between the subpopulations. In our experiment, during daily serial transfer of the metapopulations, we manipulated the mutation supply rate by creating a total of four combinations of effective population size (large and small) and migration rate (high and low, relative to the effective population size). For each of these mutation supply rates, replicate metapopulations were propagated by either the star or the well-mixed network topologies. This full-factorial design allowed us to track a total of 64 replicate metapopulations (8 replicate metapopulations × 2 topologies × 4 mutation supply rates). Migration in the star was biased towards the hub (3X IN > OUT) as previous work showed this pattern of asymmetric migration resulted in faster spread of a single beneficial mutation than symmetric bidirectional dispersal (*Chakraborty et al., 2023*).

Experiments were initiated by inoculating each subpopulation in a metapopulation with ~$10^7$ (large) or ~$10^5$ (small) colony forming units (CFU) per ml of either PA14 or PA14:*lacZ* descended from a single colony picked from an agar plate and grown overnight in liquid LB at 37°C with vigorous shaking (150 RPM). Metapopulations were transferred daily following dispersal among subpopulations (see

below) by diluting overnight cultures either 1:10$^2$ or 1:10$^4$ into fresh medium supplemented with ciprofloxacin. This transfer regime corresponds to ~6.67 (large) or ~13.28 (small) daily generations of growth ($N_t = N_0 \times 2^g$, $N_t/N_0 = 10^2$ or $10^4$, $g$ = number of generations). In our experiments, each subpopulation of a large metapopulation had a ~50 times higher effective population size ($N_e = N_0 \times g$) than that of a small metapopulation.

We constructed distinct network topologies by mixing subpopulations prior to serial transfer following the schematic shown in our previous work (Supplementary figure 4 in *Chakraborty et al., 2023*). Briefly, well-mixed networks were created by combining equal volume aliquots from all subpopulations into a common dispersal pool, diluting this mixture to the appropriate density to achieve the desired migration rate, and then mixing the dispersal pool with aliquots from each subpopulation (so-called 'self-inoculation') before transfer. Star networks, which involve bidirectional dispersal between the hub and leaves, were constructed in a similar way to the well-mixed situation only now the dispersal pool consisted of aliquots from just the leaves and aliquots from the hub (plate 2) were mixed with 'self-inoculation' samples from each leaf prior to serial transfer. For LPHM, LPLM, SPHM and SPLM ~10$^5$, ~10$^3$, ~10$^3$, ~10 CFU/ml migrants were used in addition to the self-inoculations, respectively. Further details on how each network topology and migration rate were achieved are the same as *Chakraborty et al., 2023*.

This daily transfer protocol was continued for ~100 generations, corresponding to 15 transfers (days) for the large metapopulations and 7 transfers (days) for the small metapopulations.

## Phenotypic analyses
### Measurement of fitness of the metapopulations
We measured the fitness of each evolved metapopulation every ~25 generations throughout the evolution experiment. We used growth rate ($r$) and carrying capacity ($K$) in the selective medium as proxies for fitness. During the evolution experiment, a mixture of the whole metapopulation was archived daily. We revived these metapopulation mixes saved on Days 1, 3, 7, 11, and 15 for the large metapopulations, and Days 1, 3, 5, and 7 for the small metapopulations in the selection medium by overnight growth in 1.5 ml of media in each well, in an orbital shaker (150 RPM) at 37°C along with the PA14 and PA14:LacZ ancestors. On the next day, the overnight cultures were diluted 1:1000 in fresh 200 μl selection medium in a 96-well plate and a growth curve experiment was started in a BioTek PowerWave spectrophotometer (BioTek Instruments Inc, Winooski, VT) incubated at 37°C with linear shaking for 20 s every 5 min. The optical density (OD) of each sample was measured at 600 nm after every 20 min for 24 hr until the sample reached stationary phase. Carrying capacity was measured as the highest OD reached during the 24 hr growth and growth rate was estimated as the maximum slope of the growth curve with a procedure described in another study (*Herren and Baym, 2022*). Each measurement was carried out with five biological replicates. We only revived cryo-stocks belonging to the same day together, and the measured carrying capacity and growth rate was normalized by the appropriate ancestors' growth on the same day of experiment, hence minimizing any measurement bias. Border wells of the 96-well plate were uninoculated to minimize sample loss due to evaporation.

### Measurement of fitness of the evolved isolates collected from the metapopulations
We collected 12 single isolates from each subpopulation from all the evolved metapopulations (12 isolates × 4 subpopulations × 64 metapopulations = 3072 single isolates in total). The evolved subpopulations from the last time-point in the experiment were diluted 1:10$^6$ and plated on LB-agar plates for overnight growth and inoculated at 37°C. Twelve single isolates from pre-marked positions were collected to avoid bias and were grown overnight in liquid LB medium and were cryo-preserved. The cryo-stocks for each isolate were streaked on an LB-agar plate and a single colony was grown in the selective growth medium in 24-well plates overnight at 37°C with vigorous shaking (150 RPM). The OD was measured at 600 nm in a BioTek PowerWave spectrophotometer after thoroughly mixing the overnight culture. For each isolate, carrying capacity was measured only once given the large number of total isolates for measurement. However, PA14 and PA14:*lacZ* ancestors grown 64 times each for CIP and NO-CIP environments suggest there was negligible day to day variation in carrying capacity measurements even for biological replicates (vertical red dashed lines in *Figure 5*).

## Genomic analyses

### Whole-genome sequencing

We randomly selected six replicate end-point metapopulations from each treatment combination (2 population sizes (large or small) × 2 topologies (star or well-mixed) × 6 replicates = a total of 24 metapopulations). Among the selected metapopulations, for each topology, two were from LPHM, four from LPLM, four from SPHM, and finally two were from SPLM (*Figure 1*). The rationale behind this selection was twofold: (1) LPLM and SPHM were seen to have the biggest differences between the topologies in the phenotypic analyses; and (2) the different migration rates did not influence the fitness trajectories of the metapopulations.

Both ancestral strains of *P. aeruginosa*, PA14 and PA14:*lacZ*, were also sequenced to facilitate genome assembly and to identify genetic variants arising during the evolution experiment. The four constituent subpopulations of each selected metapopulation were revived overnight from frozen stock and each metapopulation mix reconstructed by mixing the revived subpopulations in equal volumes. Genomic DNA was then extracted from each metapopulation mix for whole-genome sequencing using the QIAGEN DNeasy UltraClean 96 Microbial kit, following the manufacturer's recommended protocol. Library preparation and sequencing were performed by Genome Quebec at McGill University on the Illumina NovaSeq 6000 platform, using paired-end sequencing of 2 × 150 base-pair reads.

### Processing of genomic data and variant calling

Whole-genome sequencing of 24 metapopulations yielded a total of ~ 400 Gb of raw data, with a median depth of 3622.3-fold and an average of 98.5% genome coverage. Sequencing reads were first quality checked by generating FastQC reports using FastQC version 0.11.9 and quality trimmed using Trimmomatic version 0.39 (*Bolger et al., 2014*), with the command SLIDINGWINDOW:5:20 MINLEN:20 LEADING:5 TRAILING:5 CROP:140 HEADCROP:10. Variants were called using Breseq version 0.36.1 (*Deatherage and Barrick, 2014*) a tool specifically designed for detecting mutations in microbial genomes, with default parameters (detection limit of 5%) and -p flag for detecting polymorphisms in the sequenced metapopulations. Reads were aligned to the *P. aeruginosa* reference genome UCBPP-PA14 – assembly GCF_000014625.1 (*Winsor et al., 2016*). We subsequently discarded variants common across both ancestral strains (PA14 and PA14:*lacZ*) and all evolved populations using the gdtools module of breseq to identify only the mutations that arose over the course of the selection experiment. We found no evidence of cross-contamination in our sequenced metapopulations (each even numbered replicate had the *lacZ* insertion which was absent in the odd numbered replicates). Also, none of the metapopulations harboured any mutations in the genes that have been linked to increased mutation rates (*Oliver and Mena, 2010*; *Wiegand et al., 2008*; *Sanders et al., 2006*) in *P. aeruginosa*, so we did not have to discard any sequenced genomes from further downstream analyses. All genomic analyses were performed on Compute Canada high performance computing platform using a custom bash script for bioinformatic workflow (see data availability section for details). Breseq output files with information on genetic changes were processed further for performing statistical tests and generating plots in R statistical computing software.

## Statistical analyses

All statistical analyses were performed in R (version 4.2.2) (*R Core Team, 2022*).

We modelled the rate of change in fitness (both relative growth rate and carrying capacity) through time separately for each population size treatment using a linear mixed model (lmer) with time (*Bates et al., 2015*), migration rate and network topology as fixed factors, and a random intercept for individual replicate population to account for resampling across time (repeated measures). Time was considered to be a second order polynomial regressor in our model since the trajectories were not linear. The distribution of fitness effects among the beneficial mutants isolated from the end-point metapopulations was compared between the network topologies using a Kolmogorov–Smirnov (K–S) test paired with a resampling procedure (10,000 permutations).

To quantify gene-level parallelism in our experiment, we focused on genes reaching or exceeding a frequency of 8% and excluded synonymous mutations under the assumption they are selectively neutral (*Figure 6—figure supplement 2*, since ~97% synonymous mutations reached the frequency of ~8% or less in our experiment). Since there is no broadly accepted metric for quantifying gene-level parallelism, we use three distinct measures: variance in dispersion of Euclidean distances between

populations, Jaccard distance (Jdist = 1 − Jaccard index, which describes the likelihood that the same gene is mutated in two independent populations) and observed repeatability relative to expectation under randomness using the hypergeometric distribution (*C*-score) (sensu *Schick et al., 2022*). Briefly, a low value for dispersion, a low Jaccard distance, and a high *C*-score indicates a high degree of parallelism and vice versa.

For dispersion, we calculated the mean distance between a population and the corresponding population or network topology treatment centroid, following a principal component analysis on a Euclidean distance matrix using the vegdist function from the vegan package in R (*Oksanen et al., 2024*). The distance is measured as the population and network topology level genetic variance with larger mean dispersion signifying more divergent genetic changes and therefore less parallelism.

For the Jaccard distance measure, we calculated the Jaccard measure from the vegdist function from the vegan package in R as the dissimilarity between all pairs of populations within a treatment (effective population size or network topology) (*Schick et al., 2022*). Jaccard distance is the complement to Jaccard index and describes the likelihood that the same gene is mutated in two independent populations (here metapopulations). Jaccard distance can range 0–1: zero being two samples are exactly alike and one being two samples are completely different. We reported the difference between the network topology means (estimated difference = star − well-mixed, averaged over samples) for each effective population size treatment. For the *J*-distance, a positive difference denotes higher parallelism for well-mixed and negative difference means higher parallelism for the star topology.

Our last measure of repeatability is the *C*-score, which uses the hypergeometric distribution to calculate the deviation between the observed amount of parallelism and the expectation under random gene use (*Yeaman et al., 2018*). The magnitude of the *C*-score represents the magnitude of the deviation, larger *C*-scores signifying higher repeatability and therefore more parallelism. As for the Jaccard distance, we report the differences between the *C*-score means for each treatment combination. A positive difference denotes higher parallelism for a star, and a negative difference means higher parallelism for the well-mixed topology.

To determine the significance of each metric, we first performed a two-way ANOVA with the metric (Euclidean distance, Jaccard distance, or *C*-score) as the response variable and effective population size and network topology and their interactions as explanatory variables. We followed up any significant interaction between effective population size and network topology by measuring the difference in gene-level parallelism between the two topologies for each effective population size (significance threshold set at p < 0.05 for all three metrics) using the emmeans package from R (*Lenth, 2020*). Furthermore, to determine the significance of both Jaccard and *C*-score metrics, we performed a permutation test by randomizing population size and network topology treatment labels (number of permutations = 10,000) and calculating a null distribution of *F*-values. We complemented this analysis by calculating the probability (out of the total number of permutations) of the observed difference between the star and the well-mixed topologies being higher (*C*-score) or lower (*J*-distance) than the randomized estimated difference between the topologies, for each effective population size.

Parallelism at the gene level was defined as the proportion of populations with mutations in that gene, both globally for effective population size and network topology and within all of the four treatment combinations (*Schick et al., 2022*). To test for significance, we calculated the probability of our observed results against the null hypothesis that gene use was random, using the binomial distribution with the number of metapopulations as the number of trials, number of times a gene was mutated as the number of successes, and proportion of total metapopulations across all treatments with a mutation in that gene as the probability of success. From this, if the probability of an observation was <0.05, we considered that gene to be either effective population size or network topology treatment specific.

## Acknowledgements

This work was supported by a Natural Sciences and Engineering Research Council (NSERC) Discovery Grant to RK.

## Additional information

### Funding

| Funder | Grant reference number | Author |
|---|---|---|
| Natural Sciences and Engineering Research Council of Canada | RGPIN-2019-05622 | Rees Kassen |

The funders had no role in study design, data collection, and interpretation, or the decision to submit the work for publication.

### Author contributions

Partha Pratim Chakraborty, Conceptualization, Resources, Data curation, Formal analysis, Validation, Investigation, Visualization, Methodology, Writing – original draft, Writing – review and editing; Rees Kassen, Conceptualization, Formal analysis, Supervision, Funding acquisition, Visualization, Writing – original draft, Project administration, Writing – review and editing

### Author ORCIDs

Partha Pratim Chakraborty ⓘ https://orcid.org/0000-0002-6066-156X
Rees Kassen ⓘ https://orcid.org/0000-0002-5617-4259

### Decision letter and Author response

Decision letter https://doi.org/10.7554/eLife.107189.sa1
Author response https://doi.org/10.7554/eLife.107189.sa2

## Additional files

### Supplementary files

MDAR checklist

### Data availability

All sequence reads files are deposited in the National Center for Biotechnology Information (NCBI) Sequence Read Archive, BioProject PRJNA1401813.

The following dataset was generated:

| Author(s) | Year | Dataset title | Dataset URL | Database and Identifier |
|---|---|---|---|---|
| Chakraborty PP, Kassen R | 2026 | Accelerated evolution in networked metapopulations of *Pseudomonas aeruginosa* | https://doi.org/10.5061/dryad.sf7m0cggv | Dryad Digital Repository, 10.5061/dryad.sf7m0cggv |
| Chakraborty PP, Kassen R | 2026 | Accelerated evolution in networked metapopulations of *Pseudomonas aeruginosa* | https://www.ncbi.nlm.nih.gov/bioproject/PRJNA1401813 | NCBI BioProject, PRJNA1401813 |

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
