## [Editor Report]

This manuscript shows that certain forms of population subdivision (in particular "start" topologies) can accelerate adaptation, representing an important empirical confirmation of general predictions from evolutionary graph theory. The experiments and analysis are rigorous and provide convincing evidence for the main claims of the paper. There is some ambiguity and potential disagreement about precisely how these results relate to previous theoretical expectations, but these are matters of interpretation and we believe this article should stimulate future theory work that will further clarify the role of population structure on adaptation, particularly in situations more directly comparable to experimental settings.

---

## [Decision Letter]

**Decision letter after peer review:**

Thank you for submitting your article "Accelerated evolution in networked metapopulations of *Pseudomonas aeruginosa*" for consideration by *eLife*. Your article has been reviewed by 3 peer reviewers, and the evaluation has been overseen by a Reviewing Editor and Alan Moses as the Senior Editor.

Essential Revisions:

The reviewers all found this work to be interesting, and agree that the experiments will represent a valuable contribution to the literature. However, there is some disagreement about the relationship to prior theory, along with some concerns about methods and conclusions. The authors should consider the following essential points in a revision:

1) Please consider the comments raised by reviewer 1 in point 1.1 and 2.3. Similar points are raised by the other two reviewers. There is some concern that the acceleration of adaptation is relatively weak and only observed at a single population size, and relies on a small number of potential outlier points. These effects may be real but the authors should consider and comment on the statistical points raised by the reviewer.

2) Reviewer 1 suggests substantial theoretical or computational work. Reviewer 3 also makes related suggestions. The authors will probably feel that this is beyond the scope of the present manuscript, which is a reasonable position. However, some discussion of the general differences between their setup and earlier theory along with an intuitive description of potential implications of these differences would be useful. The authors could also comment on potential directions for future theory along these lines in the discussion, if they wish.

3) The authors should expand their analysis of parallelism as suggested by reviewers 1 and 2.

4) The point raised by reviewer 1 in point 3.1 is a significant issue that complicates the interpretation of the results. This is not reasonably addressable in a revision but this caveat should at least be acknowledged by the authors.

*Reviewer #1 (Recommendations for the authors):*

The authors carry out a short-term evolution experiment in populations with two distinct geographic structures, the "star" topology (STAR), and the "well-mixed" control (WM). The main result presented in Figures 2 and 3 is that the STAR populations adapt on average faster than the WM populations when the population size is "small" (~10^5) but not "large" (~10^6). The authors rationalize this surprising observation in terms of an increased probability of establishment of new beneficial mutations in the hub deme.

The question of how geographic structure affects evolutionary dynamics is a classic one in theoretical population genetics. The prediction of the standard theory is that subdivision should slow down adaptation (at least on short time scales) because it slows down the spread of adaptive mutations compared to well-mixed populations. However, recent developments in evolutionary graph theory (EGT) challenge some of the classical results. In particular, EGT shows that certain geographic structures, notably the star topology, can increase the probability of fixation of beneficial mutations (albeit at the cost of longer fixation times), while other structures can do the opposite.

However, the classical theory and EGT operate under different assumptions, and it is not entirely clear whether the predictions of EGT might hold at least to some extent in meta-population context. This is an interesting theoretical problem and may be important for our understanding of populations in nature. The authors chose to approach it experimentally, in the bacterium *Pseudomonas aeruginosa*, letting it adapt to sublethal concentrations of ciprofloxacin for about 100 generations. The experiments and analyses overall are reasonable, although I have some methodological concerns discussed below. The main observation, if it holds, is interesting and worth reporting.

Concerns

I have many concerns with the present version of the manuscript. While none of them individually are fatal, the paper is substantially weakened by their collective weight. Since they are all roughly equally serious, I organized them by type rather than severity.

1. Conceptual

1.1. The relevance of the main result for the field is very much in doubt for the following reasons. (a) The observed effect of the geographic structure on the rate of adaptation is rather weak (less than 10%), (b) it is observed only for one population size, (c) statistical significance is established only for carrying capacity but not growth rate, and (d) the variance in the rate of adaptation across populations may not have been estimated very accurately from just 8 replicates, which could potentially further undermine the statistical significance (see point 2.3 below). Furthermore, (e) the supporting arguments presented in Figures 4-6 are not particularly convincing (see point 2 below). This concern could be to some extent alleviated if there was a strong theoretical support for the observations. But the theory in this regime is lacking.

1.2. The absence of a theory makes the interpretation of observations difficult. The authors rely heavily on their previous paper (Ref. 26) where they showed that the star topology transiently accelerates the spread of an adapted mutant through a metapopulation. Although I take a bit of an issue with how the authors describe this effect (see below), overall that result makes sense and is almost certainly real. However, it is unclear to me whether and how this effect (which holds for an isolated beneficial mutant) would translate to the rate of adaptation in the strong clonal interference regime (which is where the experiments are carried out). Indeed, when new mutations continuously enter the population, each new most-fit class in the hub node is populated by both new mutations and migrants from peripheral nodes. New mutations likely weaken the acceleration of establishment of the most-fit class afforded by the star topology. While working out the analytical theory would most certainly constitute its own major project, it is possible to address this concern to some extent with simulations, similar to how the authors have done it in Ref. 26.

I think the simplest model to consider is the one by Desai and Fisher (Genetics 2007), though it may not be sufficient. In any case, a model like that can be simulated with selection coefficients, mutation rates and population sizes spanning the range of those realized in the experiments. I would also strongly encourage the authors to vary the number of peripheral nodes because there are good reasons to expect that it would make the effect stronger. It would then be good to generate theoretical predictions for Figures 2-5. Even if the simulations do not immediately reveal the precise mechanism of acceleration, they would (a) establish that the effect is real and is not caused by some experimental artifacts (see point 3 below); (b) show the range of effect sizes; and (c) outline the parameter range where the effect may be observable.

2. Evidence

2.1. LL 161-175. Figure 5 shows that WM populations have a larger number of mutations than STAR populations. I don't know whether this supports or contradicts the authors' hypothesis that the star topology "reduces the probability of drift loss". The authors should either explicitly discuss their expectation here or provide the results of simulations to motivate this experiment.

2.2. LL 176-198. Similar to above, it is unclear to me why we should expect more parallel evolution at the meta-population level in STAR vs WM populations. Is it possible that the results in Table 1 are just a consequence of the result in Figure 5? In other words, suppose we have two types of populations, such that populations of type 1 sample K1 mutations from some pool and populations of type 2 sample K2 mutations from the same pool. Will there be a systematic difference in parallelism between these two types if K1 ≠ K2? Does the observed difference go in the same or opposite direction from the one predicted by this simple null model?

To more intuitively understand where the difference between the parallelism statistics reported in Table 1 comes from, in addition to Figure 5, it might also be helpful to plot the frequencies of all genes that have parallel mutations + the group of all singleton mutations (for each population type separately). So, similar to Figure S3 but for all mutations and grouped by population type rather than mutated gene.

2.3. Figure 3. Both K plots (SPHM and SPLM) have one outlier each. If these outliers are removed, the blue and red curves would become more similar. Since these plots represent the main statistically significant result, I am quite concerned that significance here may be driven by these outliers. Have the authors tested whether their results remain significant if they remove the outliers? The corresponding r plots also have one outlier each. Are the same SPHM and SPLM populations responsible for both outliers in r and K? How would removing these populations change Figures 4-6?

By the same token, I am also worried about the fact that the main difference between the small STAR and WM populations is in the carrying capacity but not in growth rate, even though these populations spend most of the time growing (see point 3.1 below). This is counter to my expectations. Do the authors have a rationalization of this?

2.4 Figure 4. Why are the K values so much lower here than in Figures 2 and 3? Is it really the case that polymorphic evolved populations reach much lower carrying capacities than individual clones sampled from them? This is not impossible of course, but the effect is huge, and this surprising fact is not mentioned anywhere. Some explanation is needed.

3. Experimental issues

3.1. LL 639-648. I am quite concerned about the fact that large populations go through 6.67 divisions between transfers whereas small populations go through 13.28 divisions. This difference implies that large populations spend more time in starvation compared to small populations, which almost certainly alters the selection pressures they experience. The Petrov lab has demonstrated such effects very clearly in yeast. If the situation is similar in *P. aeruginosa*, small and large populations are probably not directly comparable. This would not necessarily undermine the central claims of the paper but would significantly affect the interpretation of Figure 6 and discussion on LL 237-240.

3.2. Figures 2 and 3. Large populations show a very consistent big jump in the first time interval (Figure 2) but this is not the case for small populations (Figure 3). Could this jump be due to physiological adaptation? If so, it may be necessary to exclude this time point from the analysis.

*Reviewer #2 (Recommendations for the authors):*

This is an exciting paper that demonstrates how the structure of spatial structure can quantitatively affect evolutionary responses to selection. The authors demonstrate that the evolutionary effects of different spatial structures are mediated by population size. These ideas have a long history in evolutionary theory, but have only undergone limited experimental testing. The combination of phenotypic and genotypic assays for the selected populations provides a more comprehensive understanding of the evolutionary responses. A more detailed description of the evolutionary responses would strengthen their conclusions and provide a broader context for evaluating their results.

This is an interesting experiment with complicated results. Looking first at the main results, in Figure 2 there are no major differences in evolutionary responses over 15 days of culturing in "large" populations, regardless of the topology. Large is 10^7 CFU per ml. The phenotypic traits measured do not directly map to competitive fitness, although growth rate is typically regarded as a good proxy. Carrying capacity is much more complicated, although it can be viewed as a measure of absolute fitness. In Figure 3, we see evidence of a difference between the two treatments (WM vs Star), with Star appearing to have a faster response than WM.

Some issues with the experimental results and analysis. 1. Adaptation is faster in larger populations, regardless of the trait and topology, than in smaller populations, but this does not appear to be mentioned in the manuscript when discussing the results (lines 130 – 198). 2. The analysis as presented appears to be in chunks (for each population size treatment), rather than an overall analysis with subsequent a priori tests. The approach in the paper could inflate p-values and prevents direct comparisons among treatments (small vs. large populations). 3. Not comparing the rate of evolutionary responses in the small and large populations necessarily avoids discussion of the rapid reduction in the rate of adaptation observed in the large populations. 4. Given the focus on phenotypic parallelism among the populations, the focus exclusively on genetic parallelism is a missed opportunity. 5. The graphs in figures 2 and 3 appear not to leverage the more sophisticated statistical analysis of populations, which is unfortunate and provides a less clear perspective of the data.

Having said that, the experiments are interesting. The results, as presented, provide an overly biased view of the actual response to selection. In addition, the complete absence of any mention of Wright's perspective on adaptation in small populations is surprising.

*Reviewer #3 (Recommendations for the authors):*

This manuscript addresses a central question in evolutionary biology: how does the spatial structure of a population influence the rate of adaptation? The authors use evolution experiments with *Pseudomonas aeruginosa* populations to test the effects of two contrasting network topologies, a well-mixed structure and a star-shaped metapopulation, under controlled migration and selection conditions. The paper builds on prior work from the group done in the single mutation limit and also prior theoretical predictions that star-like structures can increase the probability of mutant fixation by preserving beneficial mutations through a hub population.

Experimental design is robust, with multiple replicates across four metapopulations per topology and different mutation supply regimes (controlled via population size). Phenotypic measurements of fitness over time, along with sequencing at the endpoint, provide complementary evidence of rates of adaptation for star topologies. The finding that the effect disappears in high-mutation supply regimes (larger populations) also supports prior theoretical results on rates of evolution and clonal interference in these topologies (see Kuo, Hu and Carja, 2025).

Genomic analyses further strengthen the paper's claims, showing both a reduced diversity of mutations (consistent with clonal interference reduction) and a high rate of parallel evolution in the star topology. However, some key differences in fitness improvement (e.g., in growth rate) are only marginally significant, and the implications of this should be more explicitly discussed.

While this proof of concept for star topologies is important, these structures are nonetheless highly artificial. The broader applicability of the results, beyond this experimental setup is discussed briefly, but could be elaborated further. Overall, the work provides validation that population structure is not merely a complicating detail, but can play a key role in shaping evolutionary outcome.

Some recommendations I have:

1. I would cite recent work in the multi-mutational regime that is more relevant than the theoretical results the authors reference, particularly because it theoretically supports their claims and addressed the clonal interference scenario (see their Figure 6 in https://pubmed.ncbi.nlm.nih.gov/40027632/)

2. I would try to explain the connection and differences between mutation rate and migration rate in the experiments. Mutation supply rate is defined as "the product of population size, N, and mutation rate, m" (line 95), but the authors later state that supply rates are affected by "population size and migration rate" (line 127) and that is a bit confusing for a reader.

3. How are initial populations distributed, specifically, what fraction of mutant and wild-type cells are present in leaves versus hub at the start of experiments. Does it matter for outcome?

4. In cases where the p-values are marginal (e.g., growth rate differences between treatments), I would consider including effect sizes or confidence intervals.

5. Some additional discussion on how these artificial topologies nonetheless can inform on more realistic biological scenarios would be appreciated by a reader.

Thank you for allowing me to read about this great work!

---

## [Author Response]

Essential Revisions:The reviewers all found this work to be interesting, and agree that the experiments will represent a valuable contribution to the literature. However, there is some disagreement about the relationship to prior theory, along with some concerns about methods and conclusions. The authors should consider the following essential points in a revision:1) Please consider the comments raised by reviewer 1 in point 1.1 and 2.3. Similar points are raised by the other two reviewers. There is some concern that the acceleration of adaptation is relatively weak and only observed at a single population size, and relies on a small number of potential outlier points. These effects may be real but the authors should consider and comment on the statistical points raised by the reviewer.

The issue of effect sizes is an important one that we address more fulsomely in our responses below. Briefly: we have followed the suggestion of reviewer 3 and included effect sizes to be clearer about how much acceleration is happening. We find that, for small populations, stars can accelerate adaptation by (when considering *K* as our fitness proxy) as much as 1.5x that of the well-mixed, fully connected metapopulation. The effect is modest but real: dropping outliers and re-running our analyses does not change our results. Moreover, the fact that we only see acceleration at one population size (the smaller one) is consistent with expectations stemming from evolutionary graph theory. This is biologically interesting because it suggests acceleration, if and when we see it in the ‘real’ world, is a phenomenon that is associated with small populations such as in the initial stages of an infectious disease outbreak or at range edges for species expanding into new habitat. Our work therefore serves to clarify the conditions under which we expect to see topology impacting rates of adaptation.

2) Reviewer 1 suggests substantial theoretical or computational work. Reviewer 3 also makes related suggestions. The authors will probably feel that this is beyond the scope of the present manuscript, which is a reasonable position. However, some discussion of the general differences between their setup and earlier theory along with an intuitive description of potential implications of these differences would be useful. The authors could also comment on potential directions for future theory along these lines in the discussion, if they wish.

Indeed, we feel a full theoretical treatment is beyond the scope of the current manuscript. That said, we have made a number of revisions in response to the reviewer’s suggestions to better characterize existing theory and paths forward. We acknowledge, moreover, that our original explanation – reduced probability of drift loss – is probably not sufficient on its own to explain acceleration and now offer a second, not mutually exclusive, explanation: lower levels of clonal interference in stars relative to well-mixed metapopulations can decrease time to establishment for beneficial mutations. Exploring these mechanisms in a full theoretical treatment is well beyond the scope of our current manuscript. For the moment, we are content to have drawn attention to acceleration as a real, biological phenomenon grounded in experimental evidence. Hopefully our results spur further theoretical work in this exciting area.

3) The authors should expand their analysis of parallelism as suggested by reviewers 1 and 2.

We have done this by revising our mechanistic explanations to include the clonal interference effect. Please see our responses to reviewers for further explanation. We remain puzzled by reviewer 2’s request for further discussion of phenotypic parallelism as, to our thinking, it is parallelism (as measured by the variance in our measures of fitness among replicate evolving metapopulations) that is the main effect being tested in our analyses. As already explained, this effect does seem to be real and we have now quantified it by providing effect sizes.

4) The point raised by reviewer 1 in point 3.1 is a significant issue that complicates the interpretation of the results. This is not reasonably addressable in a revision but this caveat should at least be acknowledged by the authors.

The point the reviewer makes here is that large and small populations spend different amount of times in stationary phase, and so different durations of starvation. The implication is that selection might act differently in large and small populations, so comparing them may not be legitimate. While this may be the case, the available evidence suggests otherwise: the same genes associated with starvation are mutated in both conditions. Moreover, in our experience, *P. aeruginosa* populations probably spend most of a 24 hr growth cycle in stationary phase for both population sizes so any selective differences in starvation duration is not likely to be strong. Finally, we note that the time in stationary phase is the same for stars and well-mixed populations at a given population size, so contrasts in rates of adaptation between metapopulation topologies at the same population size remain legitimate.

Reviewer #1 (Recommendations for the authors):The authors carry out a short-term evolution experiment in populations with two distinct geographic structures, the "star" topology (STAR), and the "well-mixed" control (WM). The main result presented in Figures 2 and 3 is that the STAR populations adapt on average faster than the WM populations when the population size is "small" (~10^5) but not "large" (~10^6). The authors rationalize this surprising observation in terms of an increased probability of establishment of new beneficial mutations in the hub deme.The question of how geographic structure affects evolutionary dynamics is a classic one in theoretical population genetics. The prediction of the standard theory is that subdivision should slow down adaptation (at least on short time scales) because it slows down the spread of adaptive mutations compared to well-mixed populations. However, recent developments in evolutionary graph theory (EGT) challenge some of the classical results. In particular, EGT shows that certain geographic structures, notably the star topology, can increase the probability of fixation of beneficial mutations (albeit at the cost of longer fixation times), while other structures can do the opposite.However, the classical theory and EGT operate under different assumptions, and it is not entirely clear whether the predictions of EGT might hold at least to some extent in meta-population context. This is an interesting theoretical problem and may be important for our understanding of populations in nature. The authors chose to approach it experimentally, in the bacterium *Pseudomonas aeruginosa*, letting it adapt to sublethal concentrations of ciprofloxacin for about 100 generations. The experiments and analyses overall are reasonable, although I have some methodological concerns discussed below. The main observation, if it holds, is interesting and worth reporting.ConcernsI have many concerns with the present version of the manuscript. While none of them individually are fatal, the paper is substantially weakened by their collective weight. Since they are all roughly equally serious, I organized them by type rather than severity.1. Conceptual1.1. The relevance of the main result for the field is very much in doubt for the following reasons. (a) The observed effect of the geographic structure on the rate of adaptation is rather weak (less than 10%), (b) it is observed only for one population size, (c) statistical significance is established only for carrying capacity but not growth rate, and (d) the variance in the rate of adaptation across populations may not have been estimated very accurately from just 8 replicates, which could potentially further undermine the statistical significance (see point 2.3 below). Furthermore, (e) the supporting arguments presented in Figures 4-6 are not particularly convincing (see point 2 below). This concern could be to some extent alleviated if there was a strong theoretical support for the observations. But the theory in this regime is lacking.

The reviewer raises concern over whether the effect size associated with acceleration is real. We believe it is, as the treatment effect remains statistically significant even after removing outliers (more detailed responses, including re-analysis, provided in response to 2.3 below). To aid in interpretation we also now report effect sizes following the suggestion of reviewer 3. Even a modest effect size is biologically interesting because: (i) it is in the opposite direction to that expected from classic theory; and (ii) small effects on rates of adaptation in the short term of 100 generations (as in our experiment) can lead to large changes in fitness over the long run.The reviewer questions our results because acceleration is observed only in small populations. We respectfully point out: (i) acceleration at small population sizes is consistent with one of our proposed mechanisms – the avoidance of drift loss – and in tune with the predictions of EGT; (ii) our observations clarify the conditions under which acceleration is expected to be observed and inform future models. That acceleration was only observed in small populations therefore does not undermine the biological significance of our work. Rather, it points to where we might see acceleration happening in the real world. Small range edge populations, for example, may be more likely to experience acceleration than an equivalently spatially structured metapopulation in the range center.That only carrying capacity rather than growth rate shows a significant effect in our study also does not undermine our results. Both *r* and *K* are traits contributing to fitness in batch culture serial transfer experiments such as ours (Kassen 2024, Ch 1; https://www.journals.uchicago.edu/doi/epdf/10.1086/285685). Since our experiment was conducted on a 24 hr transfer schedule and the ancestor completes exponential growth in 6-8 hrs, the populations spend the bulk of their time (16-18 hrs) in stationary phase. It is therefore expected that much of the change in fitness would be observed in *K* rather than *r.*See our responses below.See our responses below.

1.2. The absence of a theory makes the interpretation of observations difficult. The authors rely heavily on their previous paper (Ref. 26) where they showed that the star topology transiently accelerates the spread of an adapted mutant through a metapopulation. Although I take a bit of an issue with how the authors describe this effect (see below), overall that result makes sense and is almost certainly real. However, it is unclear to me whether and how this effect (which holds for an isolated beneficial mutant) would translate to the rate of adaptation in the strong clonal interference regime (which is where the experiments are carried out). Indeed, when new mutations continuously enter the population, each new most-fit class in the hub node is populated by both new mutations and migrants from peripheral nodes. New mutations likely weaken the acceleration of establishment of the most-fit class afforded by the star topology. While working out the analytical theory would most certainly constitute its own major project, it is possible to address this concern to some extent with simulations, similar to how the authors have done it in Ref. 26.I think the simplest model to consider is the one by Desai and Fisher (Genetics 2007), though it may not be sufficient. In any case, a model like that can be simulated with selection coefficients, mutation rates and population sizes spanning the range of those realized in the experiments. I would also strongly encourage the authors to vary the number of peripheral nodes because there are good reasons to expect that it would make the effect stronger. It would then be good to generate theoretical predictions for Figures 2-5. Even if the simulations do not immediately reveal the precise mechanism of acceleration, they would (a) establish that the effect is real and is not caused by some experimental artifacts (see point 3 below); (b) show the range of effect sizes; and (c) outline the parameter range where the effect may be observable.

We thank the reviewer for these helpful comments and agree with them that the theory deserves more attention. As will be evident from our responses below, such an effort is well beyond the scope of our current manuscript and so we will defer a complete treatment to another paper.

That said, the reviewer’s comments have caused us to consider a second explanation in addition to the avoidance of drift loss for our results. Acceleration could also conceivably be due to reduced clonal interference in stars relative to fully-connected populations. Clonal interference will be present everywhere in a fully-connected metapopulation. In a star, however, leaf subpopulations do not exchange migrants with each other (only with the hub) so clonal interference will be lower in the leaves relative to that observed in the fully-connected case. This means a beneficial mutation arising in the leaf of the star – especially when population sizes are small – could fix more rapidly and so be more likely to seed the hub (and as a consequence the rest of the population), even if the likelihood of drift loss doesn’t change. If reduced clonal interference is the cause of acceleration, then we would expect to see higher rates of acceleration with lower population sizes, as we do, because clonal interference will be lower on average in leaves at small population sizes.

We note that it is conceivable that these two explanations – reduced clonal interference in the leaves and reduced probability of drift loss in the hub – could work in concert. A more sophisticated theoretical study could be helpful in teasing these effects apart. That said, we have now amended our text to reflect this alternative interpretation and highlighted the need for theory to help decide whether these mechanisms are competing or complementary. Doing so will also shed light, as the reviewer suggests, on the range of effect sizes and parameter space in which acceleration occurs.

2. Evidence2.1. LL 161-175. Figure 5 shows that WM populations have a larger number of mutations than STAR populations. I don't know whether this supports or contradicts the authors' hypothesis that the star topology "reduces the probability of drift loss". The authors should either explicitly discuss their expectation here or provide the results of simulations to motivate this experiment.

The difference in segregating mutations is consistent with the expectation that migration in the WM treatment increases clonal interference relative to that experienced in the STAR treatment. This result lends support to the idea that acceleration observed in STARS is associated with the enrichment of early-arising beneficial mutations that avoid drift loss and reduced clonal interference, as noted above (see response to 1.2). We have amended the text appropriately.

2.2. LL 176-198. Similar to above, it is unclear to me why we should expect more parallel evolution at the meta-population level in STAR vs WM populations. Is it possible that the results in Table 1 are just a consequence of the result in Figure 5?

The argument relies on the idea that early arising beneficial mutations spread more rapidly in STAR than WM because of the combination of reduced probability of drift loss and, we suspect, reduced clonal interference in stars. The result is that these beneficial mutants rise to higher frequency more rapidly in stars compared to well mixed populations, a result that is also consistent with a new analysis, suggested by the reviewer in the comment below and reported, now, in Figure S6.

In other words, suppose we have two types of populations, such that populations of type 1 sample K1 mutations from some pool and populations of type 2 sample K2 mutations from the same pool. Will there be a systematic difference in parallelism between these two types if K1 ≠ K2? Does the observed difference go in the same or opposite direction from the one predicted by this simple null model?

The argument we have outlined above does not rely on there being a difference in the pool of mutations being sampled. We note further that there does not appear to be any mutations specific to any of the topology treatments at a given population size, hence K1⋍K2 (see Figure 6).

To more intuitively understand where the difference between the parallelism statistics reported in Table 1 comes from, in addition to Figure 5, it might also be helpful to plot the frequencies of all genes that have parallel mutations + the group of all singleton mutations (for each population type separately). So, similar to Figure S3 but for all mutations and grouped by population type rather than mutated gene.

We have followed the reviewer’s suggestion and provide the plot in Author response image 1 and as a supplementary figure (S6) referred to in the main text on line 221. Mutations on average reach a higher frequency in large populations compared to small populations. More interesting, arguably, is that mutations in the small star are on average at a higher frequency than they are in the small well-mixed population. Parallelism is highest in the small STAR treatment, consistent with the idea that acceleration is associated with the enrichment of one or a few high fitness mutants.

**Author response image 1. sa2fig1:** 

2.3. Figure 3. Both K plots (SPHM and SPLM) have one outlier each. If these outliers are removed, the blue and red curves would become more similar. Since these plots represent the main statistically significant result, I am quite concerned that significance here may be driven by these outliers. Have the authors tested whether their results remain significant if they remove the outliers?

We identified which data points are outliers easily using boxplots (the points that are not the part of the box and whiskers ie. out of the 25th and 75th quartile range). Specifically, STAR metapopulation ID 47 (small population size, low migration rate, metapopulation replicate 7) and WM metapopulation ID 56 (small population size, high migration rate, metapopulation replicate 8) were identified as outliers, as evident from Author response image 2. To be conservative, we filtered out these datapoints from the analyses at all time-points (even though they are not outliers at all time-points) and redid our analyses. Results are presented below.

**Author response table 1. sa2table1:** Analysis of Deviance Table (Type II Wald chisquare tests).

Response: fold_change_k				
	Chisq	Df	Pr(>Chisq)	
poly(Day, 2)	620.6681	2	< 2e-16	***
Treatment	5.1939	1	0.02267	*
Mig_rate	0.4826	1	0.48726	
poly(Day, 2):Treatment	6.4263	2	0.04023	*
poly(Day, 2):Mig_rate	3.3172	2	0.19040	
Treatment:Mig_rate	0.3448	1	0.55709	
poly(Day, 2):Treatment:Mig_rate	1.4588	2	0.48219	

Signif. codes: 0 ‘***’ 0.001 ‘**’ 0.01 ‘*’ 0.05 ‘.’ 0.1 ‘ ’ 1.

Reassuringly, the exclusion of the outliers from the dataset does not change the main results. We have now noted this in the main text (lines 144-5 and in Supplementary Figure S4, which also includes the results of the statistical analysis in the caption).

The corresponding r plots also have one outlier each. Are the same SPHM and SPLM populations responsible for both outliers in r and K?

The only outlier for the r plots were small metapopulation ID 47, the same as the K plots.

**Author response image 3. sa2fig3:** 

**Author response table 2. sa2table2:** Analysis of Deviance Table (Type II Wald chisquare tests).

Response: fold_change_k				
	Chisq	Df	Pr(>Chisq)	
poly(Day, 2)	416.6435	2	< 2e-16	***
Treatment	3.2106	1	0.07316	
Mig_rate	0.0912	1	0.76264	
poly(Day, 2):Treatment	1.4415	2	0.48639	
poly(Day, 2):Mig_rate	0.1835	2	0.91236	
Treatment:Mig_rate	0.0284	1	0.86612	
poly(Day, 2):Treatment:Mig_rate	2.0834	2	0.35285	

Signif. codes: 0 ‘***’ 0.001 ‘**’ 0.01 ‘*’ 0.05 ‘.’ 0.1 ‘ ’ 1.

Exclusion of the datapoint does not change the result.

How would removing these populations change Figures 4-6?

We have now performed this analysis. The results are presented below.

**Author response image 4. sa2fig4:** 

Reassuringly, the exclusion of outliers does not change the results.Figure 5 and 6 do not include the Star “outlier” that shows a high K value in Figure 3 SPHM and SPLM.

Analysis of Variance Table

**Author response table 3. sa2table3:** Response: mut_count.

	Df	Sum Sq	Mean Sq	F value	Pr(>F)
Pop_size	1	130.73	130.731	2.6323	0.1212
Treatment	1	160.98	160.975	3.2412	0.0877
Pop_size:Treatment	1	0.57	0.573	0.0115	0.9156
Residuals	19	943.93	49.665		

Signif. codes: 0 ‘***’ 0.001 ‘**’ 0.01 ‘*’ 0.05 ‘.’ 0.1 ‘ ’ 1.

**Author response table 4. sa2table4:** Pop_size = large.

contrast estimate	SE	df	t ratio	p value
AMP – WM	-5.00	4.07	-1.229	0.2341

**Author response table 5. sa2table5:** Pop_size = small.

contrast estimate	SE	df	t ratio	p value
AMP – WM	5.63	4.27	-1.320	0.2025

Results of Figure 5 did not change after exclusion of the outlier P5HMW8.

By the same token, I am also worried about the fact that the main difference between the small STAR and WM populations is in the carrying capacity but not in growth rate, even though these populations spend most of the time growing (see point 3.1 below). This is counter to my expectations. Do the authors have a rationalization of this?

As mentioned above in response to point 1.1, the phenotypic target of selection is of less interest than the fitness differences they confer. That said, two additional points are relevant:

It is not, in fact, true that these populations spend most of their time growing. In our experience, these populations typically reach stationary phase (ie, carrying capacity) after 6-8 hrs of growth. On a 24-hour transfer protocol the bulk of their time is spent near carrying capacity, which could explain why carrying capacity showed the strongest response to selection.We note further that our estimates of the effect for *r* are close to formally significant at the p<0.05 level, even after removing the outlier strains. It is possible that since our ancestral strain (PA14) has a propensity to produce biofilms (and we find putative mutations upregulating biofilm formation in the c-GMP pathway in our evolved metapopulations), our estimates of OD in exponential phase are more variable than they might otherwise be, which could explain why the statistical tests are close to but not quite significant. We note further that the direction of change in *r* and *K* is positively correlated (and quite strong, being above 0.82 in all cases, suggesting with a bit more replication we would see a statistically significant effect on *r*, in addition to the convincing effect we see for *K*).

**Author response image 5. sa2fig5:** 

The facet labels are different days. This plot is only for the small population size treatments.

2.4 Figure 4. Why are the K values so much lower here than in Figures 2 and 3? Is it really the case that polymorphic evolved populations reach much lower carrying capacities than individual clones sampled from them? This is not impossible of course, but the effect is huge, and this surprising fact is not mentioned anywhere. Some explanation is needed.

The reason has to do with the difference between relative and absolute values of *K*. In figure 3, we report K relative to the ancestor (y axis; relative K) whereas in figure 4 we show absolute K of the evolved isolates. The bins used to construct in the histogram using absolute K better reflect the changes in the distribution of *K* observed during our experiment. When we re-analyze the data using the distribution of relative *K* values, our results are, as shown in Author response image 6, reassuringly not changed: the differences between the distributions are still significant.

**Author response image 6. sa2fig6:** 

3. Experimental issues3.1. LL 639-648. I am quite concerned about the fact that large populations go through 6.67 divisions between transfers whereas small populations go through 13.28 divisions. This difference implies that large populations spend more time in starvation compared to small populations, which almost certainly alters the selection pressures they experience. The Petrov lab has demonstrated such effects very clearly in yeast. If the situation is similar in *P. aeruginosa*, small and large populations are probably not directly comparable.

We agree (and have noted the lack of comparability in our response to reviewer 2, who asked for more discussion on how population size impacts rates of adaptation). As our main focus is in comparing the effect of topology on rates of adaptation we elected to focus attention on contrasts between topology at a given population size. Thus, even if the effects of starvation were more severe in the large populations than in the small ones, they would have been experienced in the same way by both topologies at a given population size. Additionally, a recent study shows that in LB (the rich laboratory media that we use in our study) *Pseudomonas aeruginosa* can maintain viable cells from 10^10 to 10^11 cfu/ml from 24 to 72 hours (https://pmc.ncbi.nlm.nih.gov/articles/PMC11504098/). Since, we bottlenecked every 24 hours, we can rule out any confounding effect of prolonged starvation on PA14 evolved in the large metapopulations.

This would not necessarily undermine the central claims of the paper but would significantly affect the interpretation of Figure 6 and discussion on LL 237-240.

It is not entirely clear to us how different times spent in starvation mode at different population sizes would impact the interpretation of Figure 6. While there are some clear examples that a handful of mutations are differentially selected in large and small populations, these are not obviously associated with adaptation to starvation conditions. PA14 is expected to adapt to starvation by biofilm dispersal, stringent response, metabolic shifts and changes in gene expression. This would mean decreased levels of c-di-GMP, higher (p)ppGpp alarmone and virulence factors etc. We uncover various genes that modulate the levels of c-di-GMP and virulence factors mutated in both large and small metapopulations (wspF/R/A, morA, pilB/D/F) without any population size specific occurrence. This reinforces that different lengths of time spent in starvation likely does not confound the interpretation of our results.

3.2. Figures 2 and 3. Large populations show a very consistent big jump in the first time interval (Figure 2) but this is not the case for small populations (Figure 3). Could this jump be due to physiological adaptation? If so, it may be necessary to exclude this time point from the analysis.

We disagree that excluding this early time point from the analysis is necessary. Our main interest is in contrasting the effects of topology at a given population size, so the comparison of fitness dynamics between large and small populations is not directly relevant to our main conclusions.

That said, while we cannot definitively rule out physiological adaptation as the cause of the jump in fitness in large populations (because we haven’t sequenced this early time point) we note that this explanation seems unlikely. Large populations will have more access to large-effect beneficial mutations early in the experiment than small populations, which readily explains this early jump in fitness. Consistent with this, we show that the large metapopulations in our experiment often fix gyrA mutations that previous work (Melnyk mSphere 2: e00158-17, 2017) showed results in large changes in resistance and, based on our previous measurements, are more advantageous than other ciprofloxacin resistant mutations at ~40ng/ml ciprofloxacin.

Reviewer #2 (Recommendations for the authors):This is an exciting paper that demonstrates how the structure of spatial structure can quantitatively affect evolutionary responses to selection. The authors demonstrate that the evolutionary effects of different spatial structures are mediated by population size. These ideas have a long history in evolutionary theory, but have only undergone limited experimental testing. The combination of phenotypic and genotypic assays for the selected populations provides a more comprehensive understanding of the evolutionary responses. A more detailed description of the evolutionary responses would strengthen their conclusions and provide a broader context for evaluating their results.This is an interesting experiment with complicated results. Looking first at the main results, in Figure 2 there are no major differences in evolutionary responses over 15 days of culturing in "large" populations, regardless of the topology. Large is 10^7 CFU per ml. The phenotypic traits measured do not directly map to competitive fitness, although growth rate is typically regarded as a good proxy. Carrying capacity is much more complicated, although it can be viewed as a measure of absolute fitness. In Figure 3, we see evidence of a difference between the two treatments (WM vs Star), with Star appearing to have a faster response than WM.

See our response to reviewer #1 regarding the relationship between growth rate and carrying capacity as components of fitness in the context of our experiment.

Some issues with the experimental results and analysis.1. Adaptation is faster in larger populations, regardless of the trait and topology, than in smaller populations, but this does not appear to be mentioned in the manuscript when discussing the results (lines 130 – 198).

Thank you for noting this. We chose to emphasize the role of topology rather than population size as the former is less well studied and understood than the latter. Moreover, for reasons articulated in response to point 2 below, a formal comparison across population size is probably not appropriate. That said, we now note the population size effect on lines 145-146.

2. The analysis as presented appears to be in chunks (for each population size treatment), rather than an overall analysis with subsequent a priori tests. The approach in the paper could inflate p-values and prevents direct comparisons among treatments (small vs. large populations).

See our response to point 1 above regarding our focus on topology rather than population size. We note that including population size in a complete analysis reveals a large main effect of population size, as expected, and, reassuringly, does not otherwise change our results. The results of an omnibus test including ‘population size’ as a fixed factor are shown below.

**Author response table 6. sa2table6:** Analysis of Deviance Table (Type II Wald chisquare tests).

Response: fold_change_k				
	Chisq	Df	Pr(>Chisq)	
poly(Day, 2)	391.6967	2	< 2.2e-16	***
Pop_size	1.0132	1	0.3141308	
Treatment	0.8521	1	0.3559552	
Mig_rate	0.0032	1	0.9547399	
poly(Day, 2):Pop_size	18.1228	2	0.0001161	***
poly(Day, 2):Treatment	1.2515	2	0.5348471	
Pop_size:Treatment	1.5562	1	0.2122273	
poly(Day, 2):Mig_rate	1.6337	2	0.4418171	
Pop_size:Mig_rate	0.7332	1	0.3918523	
Treatment:Mig_rate	0.0139	1	0.9062678	
poly(Day, 2):Pop_size:Treatment	0.9580	2	0.6194079	
poly(Day, 2):Pop_size:Mig_rate	0.5315	2	0.7666358	
poly(Day, 2):Treatment:Mig_rate	0.2008	2	0.9044948	
Pop_size:Treatment:Mig_rate	0.0384	1	0.8446889	
poly(Day, 2):Pop_size:Treatment:Mig_rate	0.2420	2	0.8860213	

Signif. codes: 0 ‘***’ 0.001 ‘**’ 0.01 ‘*’ 0.05 ‘.’ 0.1 ‘ ’ 1.

**Author response table 7. sa2table7:** Analysis of Deviance Table (Type II Wald chisquare tests).

Response: fold_change_k				
	Chisq	Df	Pr(>Chisq)	
poly(Day, 2)	476.1801	2	< 2.2e-16	***
Pop_size	7.9828	1	0.004722	**
Treatment	4.7775	1	0.028833	*
Mig_rate	0.1931	1	0.660310	
poly(Day, 2):Pop_size	39.6929	2	2.403e-09	***
poly(Day, 2):Treatment	0.4113	2	0.814130	
Pop_size:Treatment	1.9869	1	0.158663	
poly(Day, 2):Mig_rate	0.1203	2	0.941628	
Pop_size:Mig_rate	0.2862	1	0.592641	
Treatment:Mig_rate	0.1344	1	0.713942	
poly(Day, 2):Pop_size:Treatment	3.6815	2	0.158698	
poly(Day, 2):Pop_size:Mig_rate	0.6942	2	0.706739	
poly(Day, 2):Treatment:Mig_rate	0.0364	2	0.981947	
Pop_size:Treatment:Mig_rate	0.3081	1	0.578840	
poly(Day, 2):Pop_size:Treatment:Mig_rate	0.6055	2	0.738798	

Signif. codes: 0 ‘***’ 0.001 ‘**’ 0.01 ‘*’ 0.05 ‘.’ 0.1 ‘ ’ 1.

We caution, however, that this test is not strictly appropriate because the values of the variable Day (1,3,7,11,15) in large metapopulations do not correspond to the same number of generations as they do in the small metapopulations (1,3,5,7) because of the difference in dilution factors used to create large and small populations. For this reason, and because our main focus is the effect of topology on rates of adaptation at a given population size, we now mention the population size effect (lines 145-6) in the manuscript but report the previous analyses (in chunks) in the text.

3. Not comparing the rate of evolutionary responses in the small and large populations necessarily avoids discussion of the rapid reduction in the rate of adaptation observed in the large populations.

We caution, however, that this test is not strictly appropriate because the values of the variable Day (1,3,7,11,15) in large metapopulations do not correspond to the same number of generations as they do in the small metapopulations (1,3,5,7) because of the difference in dilution factors used to create large and small populations. For this reason, and because our main focus is the effect of topology on rates of adaptation at a given population size, we now mention the population size effect (lines 145-6) in the manuscript but report the previous analyses (in chunks) in the text.

4. Given the focus on phenotypic parallelism among the populations, the focus exclusively on genetic parallelism is a missed opportunity.

This comment is not clear to us. What, specifically, does the reviewer mean by phenotypic parallelism, and what opportunity is being missed? Phenotypic parallelism at the level of fitness is almost always very high (Figure 3, Bailey *et al.* 2015, https://academic.oup.com/mbe/article/32/6/1436/1070729) and our results are no exception. We focus on gene-level parallelism because it is central to understanding the genetic causes and repeatability of adaptation (see Bailey *et al.* 2017, https://doi.org/10.1002/bies.201600176).

5. The graphs in figures 2 and 3 appear not to leverage the more sophisticated statistical analysis of populations, which is unfortunate and provides a less clear perspective of the data.

This comment is also unclear. Our results very clearly show a statistically significant difference in rates of adaptation across treatments so it is unclear what additional tests might accomplish. However, we have now shown even with population size as an additional fixed factor the effect of network topology remains statistically significant (p<0.05). We welcome any suggestions for a specific test the reviewer thinks is more appropriate to ascertain our interpretation of the data.

Having said that, the experiments are interesting. The results, as presented, provide an overly biased view of the actual response to selection. In addition, the complete absence of any mention of Wright's perspective on adaptation in small populations is surprising.

We are pleased the reviewer found our work interesting. Thank you for pointing out the relevance of our work to Wright’s shifting landscape models of adaptation. We have now included a brief description of how our work relates to this model in the discussion.

Reviewer #3 (Recommendations for the authors):This manuscript addresses a central question in evolutionary biology: how does the spatial structure of a population influence the rate of adaptation? The authors use evolution experiments with *Pseudomonas aeruginosa* populations to test the effects of two contrasting network topologies, a well-mixed structure and a star-shaped metapopulation, under controlled migration and selection conditions. The paper builds on prior work from the group done in the single mutation limit and also prior theoretical predictions that star-like structures can increase the probability of mutant fixation by preserving beneficial mutations through a hub population.Experimental design is robust, with multiple replicates across four metapopulations per topology and different mutation supply regimes (controlled via population size). Phenotypic measurements of fitness over time, along with sequencing at the endpoint, provide complementary evidence of rates of adaptation for star topologies. The finding that the effect disappears in high-mutation supply regimes (larger populations) also supports prior theoretical results on rates of evolution and clonal interference in these topologies (see Kuo, Hu and Carja, 2025).Genomic analyses further strengthen the paper's claims, showing both a reduced diversity of mutations (consistent with clonal interference reduction) and a high rate of parallel evolution in the star topology. However, some key differences in fitness improvement (e.g., in growth rate) are only marginally significant, and the implications of this should be more explicitly discussed.

Please see our response to reviewer 1 regarding effect sizes and the impact of outliers on our results.

While this proof of concept for star topologies is important, these structures are nonetheless highly artificial. The broader applicability of the results, beyond this experimental setup is discussed briefly, but could be elaborated further. Overall, the work provides validation that population structure is not merely a complicating detail, but can play a key role in shaping evolutionary outcome.

Thank you for this generous comment. We note that we mention in the discussion the relevance of our work to understanding both fundamental models of evolution (penultimate paragraph) and real world scenarios involving pathogen spread and range size evolution (last paragraph).

Some recommendations I have:1. I would cite recent work in the multi-mutational regime that is more relevant than the theoretical results the authors reference, particularly because it theoretically supports their claims and addressed the clonal interference scenario (see their Figure 6 in https://pubmed.ncbi.nlm.nih.gov/40027632/)

We thank the reviewer for pointing out this citation and have made the necessary revisions (line 163).

2. I would try to explain the connection and differences between mutation rate and migration rate in the experiments. Mutation supply rate is defined as "the product of population size, N, and mutation rate, m" (line 95), but the authors later state that supply rates are affected by "population size and migration rate" (line 127) and that is a bit confusing for a reader.

Thank you for picking up on this. We have revised the text on line 100 to read, “…when mutation supply rates in each subpopulation (which is *NU*, as defined above times *m,* the migration rate into a subpopulation) are < 1 …”

3. How are initial populations distributed, specifically, what fraction of mutant and wild-type cells are present in leaves versus hub at the start of experiments. Does it matter for outcome?

All populations are isogenic at the start of the experiment. All genetic variation arises de novo through mutation.

4. In cases where the p-values are marginal (e.g., growth rate differences between treatments), I would consider including effect sizes or confidence intervals.

Thank you for this suggestion. We now include effect sizes in our comparisons where p-values are marginal. Below are the results calculated for Cohen’s *d*:

**Author response table 8. sa2table8:** Small population size, *r*.

	contrast estimate	SE	df	T ratio	P value
AMP – WM	0.0787	0.0398	47.7	1.978	0.0538

Results are averaged over the levels of: Mig_rate.

Degrees-of-freedom method: kenward-roger.

**Author response table 9. sa2table9:** eff_size(emm_T, σ = σ(lmer.r.small.pop), edf = edf).

	contrast effect.size	SE	df	Lower CL	Upper CL
AMP – WM	0.929	0.479	47.7	0.0349	1.89

Results are averaged over the levels of: Mig_rate σ used for effect sizes: 0.08469.

Degrees-of-freedom method: inherited from kenward-roger when re-gridding.

Confidence level used: 0.95.

**Author response table 10. sa2table10:** Small population size, *K*.

	contrast estimate	SE	df	t.ratio	p.value
AMP – WM	0.709	0.206	51.2	3.441	0.0012

Results are averaged over the levels of: Mig_rate.

Degrees-of-freedom method: kenward-roger.

**Author response table 11. sa2table11:** 

	contrast effect.size	SE	df	lower.CL	upper.CL
AMP – WM	1.53	0.468	51.2	0.586	2.47

Results are averaged over the levels of: Mig_rate σ used for effect sizes: 0.4649.

Degrees-of-freedom method: inherited from kenward-roger when re-gridding Confidence level used: 0.95.

5. Some additional discussion on how these artificial topologies nonetheless can inform on more realistic biological scenarios would be appreciated by a reader.

Thank you for this suggestion. We note the final paragraph of our manuscript is devoted to addressing this question. For the moment, we know too little about the topological structure of most natural populations to say much of anything. Our hope is that our experiments will spur further research into this important and fascinating subject.